# A convex optimization formulation for multivariate regression

**Yunzhang Zhu**
Department of Statistics
Ohio State University
Columbus, OH 43210
zhu.219@osu.edu

## Abstract

Multivariate regression (or multi-task learning) concerns the task of predicting the value of multiple responses from a set of covariates. In this article, we propose a convex optimization formulation for high-dimensional multivariate linear regression under a general error covariance structure. The main difficulty with simultaneous estimation of the regression coefficients and the error covariance matrix lies in the fact that the negative log-likelihood function is not convex. To overcome this difficulty, a new parameterization is proposed, under which the negative log-likelihood function is proved to be convex. For faster computation, two other alternative loss functions are also considered, and proved to be convex under the proposed parameterization. This new parameterization is also useful for covariate-adjusted Gaussian graphical modeling in which the inverse of the error covariance matrix is of interest. A joint non-asymptotic analysis of the regression coefficients and the error covariance matrix is carried out under the new parameterization. In particular, we show that the proposed method recovers the oracle estimator under sharp scaling conditions, and rates of convergence in terms of vector $\ell_\infty$ norm are also established. Empirically, the proposed methods outperform existing high-dimensional multivariate linear regression methods that are based on either minimizing certain non-convex criteria or certain two-step procedures.

## 1 Introduction

*Multivariate linear regression* concerns the task of predicting the value of one or multiple responses from a set of predictors. It is commonly used in applications where more than one responses are recorded for each sample [Reinsel and Velu, 1998]. For example, in longitudinal data analysis, the response variable is often measured on multiple different time points for the same subject in the experiment, thus resulting in multiple responses for each subject. In many prediction problems, one may be interested in predicting multiple quantities of interests. This also results in a regression problem with multiple responses.

High dimensional multivariate regression has received significant attention recently in the literature. Some earlier works mainly focus on estimating the regression coefficients while ignoring the error covariance structure. Certain regularizations are often used to either get a regularized fit or/and to reduce the model dimension. For example, some classical works [Anderson, 1951, Izenman, 1975, Reinsel and Velu, 1998] impose a low-rank constraint on the regression coefficients, whereas more recent works Turlach et al. [2005], Yuan et al. [2007], Peng et al. [2010] adopt other forms of continuous regularization to achieve low-rank or/and group-sparsity structure for the regression coefficient estimates. Theoretical justifications for the use of group sparsity penalties are provided in Obozinski et al. [2011].

There have been a few recent methods for multivariate linear regression that accounts for general error covariance structure. For instance, Rothman et al. [2010] propose to impose an $\ell_1$ penalty on both the regression coefficients and the error precision matrix to obtain sparse estimates. Lee and Liu [2012], Sofer et al. [2012] consider somewhat different sparsity-inducing penalties and provide some theoretical justifications of their procedures. More recently, She et al. [2015] study a similar model under the context of dynamical network analysis. However, all those methods involve non-convex criteria, and thus could be potentially unstable, and less amenable to theoretical analysis. Yin and Li [2013] propose a two-stage penalized procedure that estimates the regression coefficients at the first step and then estimate the precision matrix at the second step, which again can be viewed as an approximate solution to the regularized maximum likelihood method. Liu et al. [2014] formulate the problem into a convex optimization problem, but the error covariance they considered is constrained to be a diagonal matrix. More recently, Wang [2015] proposed a method to incorporate response dependence into the model via considering joint multivariate normal model, which is essentially a pseudolikelihood approach. However, the empirical criterion he considered may not be convex. Moreover, Molstad and Rothman [2016] make the same joint multivariate normal assumption, but make no explicit assumption on the regression coefficient and the error covariance, which is somewhat different from most existing works in the literature and thus is not directly comparable to our work.

In this article, we propose a convex optimization formulation to simultaneously estimate the regression coefficient matrix $C$ and the error precision matrix $\Omega$ in a multivariate linear regression model. The main issue with most of aforementioned existing approaches is that the negative log-likelihood function is not convex in the default parameteriztion $(C, \Omega)$. To circumvent this issue, we propose a new parameterization $(B, \Omega)$ under which the negative log-likelihood function is convex. Moreover, we show that the new parameterization $B$ retains the same rank and row-sparsity as the old parameterization $C$. Consequently, similar structured estimate for $C$ can be obtained by imposing appropriate regularization functions on $B$. Moreover, we show that the proposed parameterization not only works with negative loglikelihood loss function, but also works for two other alternative loss functions proposed recently by Zhang and Zou [2014], Lee and Hastie [2015]. These two alternative loss functions are more amenable to computation compared to the negative log-likelihood loss. The proposed convex formulation is in contrast with most existing high-dimensional multivariate linear regression methodologies, most of which either focus only on estimating the regression coefficients alone, or estimating the regression coefficients and the precision matrix alternately by solving a nonconvex optimization problem.

Computationally, we propose to use a *proximal Newton algorithm* [see, e.g., Lee et al., 2014] to solve the associated optimization problem for regularized negative log-likelihood loss formulation under the proposed parameterization. The proximal Newton algorithm, however, may not be scalable to large-scale problems. This is mainly caused by the $\log \det(\cdot)$ in the Gaussian log-likelihood function. For large-scale problems, we propose to use the aforementioned alternative loss functions, which can be much easier to deal with using accelerated proximal gradient methods [see, e.g., Beck and Teboulle, 2009]. Theoretically, we show that under optimal scaling conditions, and a minimum signal strength condition, the proposed method when coupled with sparsity-inducing nonconvex penalties [see, e.g., Fan and Li, 2001, Zhang, 2010, Shen et al., 2012, among others] can recover the oracle estimator with probability tending to 1. Moreover, rates of convergence of both the regression coefficient matrix and precision matrix in vector $\ell_\infty$ norm are derived, and compares favorably against existing non-asymptotic results under similar conditions.

In summary, the key contributions of this article are as follows. First, we develop a convex optimization formulation for estimating both the regression coefficients and the error covariance matrix in high-dimensional multivariate linear regression under three different loss functions. Unlike previous works that alternately estimates the regression coefficients and the error covariance matrix, the proposed method minimizes a joint convex loss function subject to regularization, and thus will not suffer from the potential local minimum as most existing methods. Second, new theoretical results are obtained for the joint estimation approach. In particular, sparse recovery results and optimal parameter estimation bound are derived under minimum conditions.

The rest of the article is structured as follows. We introduce the proposed new parameterization and the convex optimization formulation in Section 2. In section 3 we provide computational algorithms to solve the associated optimization problems. Sparse recovery results and parameter estimation error bound are derived in Section 4. Section 5 includes numerical results comparing our methods to competing methods, and we close with some remarks in Section 6.

## 2 Models

Multivariate regression concerns a regression of a $q$-dimensional response vector $y$ on a $p$-dimensional predictor vector $x$. Specifically, let $y_i = (y_{i1}, \cdots, y_{iq})^\top$ be the response vector of dimension $q$ and $x_i = (x_{i1}, \cdots, x_{ip})^\top$ be a $p$-dimensional predictor for the $i$-th sample, $i = 1, \cdots, n$. Multivariate linear regression models assume that the response vector $y_i$ relates to the covariates $x_i$ linearly

$$y_i = C^\top x_i + \epsilon_i, \quad i = 1, \cdots, n, \tag{1}$$

where $C \in \mathbb{R}^{p \times q}$ denotes the regression coefficient matrix and $\epsilon_1, \ldots, \epsilon_n \in \mathbb{R}^q$ denote the IID error vectors, which are assumed to follow a multivariate normal distribution with a general covariance matrix $\Sigma$: $\epsilon_i \sim N(0, \Sigma), i = 1, \cdots, n$. If we stack $y_i$'s, $x_i$'s, and $\epsilon_i$'s into three matrices $Y \in \mathbb{R}^{n \times q}$, $X \in \mathbb{R}^{n \times p}$, and $E \in \mathbb{R}^{n \times q}$ with $y_i^\top$, $x_i^\top$, and $\epsilon_i^\top$ being the $i$-th row of $Y$, $X$ and $E$, then we can express the above model (1) into a matrix form

$$Y = XC + E. \tag{2}$$

Let $\Omega = \Sigma^{-1}$ denote the precision matrix, whose entries encode the covariate-adjusted conditional dependence among components of the response [Yin and Li, 2013]. The negative log-likelihood function of $(C, \Omega)$ can be expressed up to a constant as

$$L_n(C, \Omega) = \frac{1}{2} \operatorname{Tr} \left\{ \Omega (Y - XC)^\top (Y - XC) \right\} - \frac{n}{2} \log \det(\Omega). \tag{3}$$

When $X$ and $Y - X\widehat{C}^{\text{MLE}}$ have full column rank, the *maximum likelihood estimator* (MLE) for $C$ and $\Omega$ has the following closed form formulas

$$\widehat{C}^{\text{MLE}} = (X^\top X)^{-1} X^\top Y, \quad \widehat{\Omega}^{\text{MLE}} = \left\{ \frac{1}{n} (Y - X\widehat{C}^{\text{MLE}})^\top (Y - X\widehat{C}^{\text{MLE}}) \right\}^{-1}. \tag{4}$$

When the problem dimension exceeds the sample size, regularization are often employed. Due to (possible) non-convexity of the negative log-likelihood function $L_n(C, \Omega)$ in (3), when regularization functions are added for $C$ and/or $\Omega$ in the high-dimensional case, the estimates based on solving the resulting optimization problem may be unstable. To address this non-convexity issue, existing works [e.g., Rothman et al., 2010, Lee and Liu, 2012, Sofer et al., 2012, among others] obtain estimates of $C$ and $\Omega$ in an alternating fashion. One downside of such an alternating strategy is that it may not be stable. For example, when using the R package provided by [Rothman et al., 2008], if the penalty parameter for the regression coefficient matrix is too small, $p > n$, and $\Omega$ is not penalized, then minimizing over $C$ sometimes leads to a perfect fit, that is $Y = XC$. Then minimizing over $\Omega$ is not a well defined problem since the minimizer is at the infinity. This will actually cause the R program to terminate in this case. In this article, we propose to tackle this issue by proposing a new parameterizations under which the negative log-likelihood function is convex.

Throughout, we shall use $C^0$, $\Sigma^0$, $\Omega^0$ to denote the true regression coefficient matrix, true error covariance matrix, and true error precision matrix, respectively.

### 2.1 A new parameterization

We consider a new parameterization for the multivariate linear regression model, which results in convex negative log-likelihood function under the new parameterization. We also demonstrate that the proposed parameterization can be used for estimating both the regression coefficients and the error covariance matrix.

To overcome the aforementioned non-convexity issue, we consider a new parameterization: $B = C\Omega$. Then it can be easily verified that the negative log-likelihood in this new parameterization is proportional to

$$L_n(B, \Omega) = -\frac{1}{2} \log \det(\Omega) + \frac{1}{2n} \operatorname{Tr} \left[ \Omega^{-1} (Y\Omega - XB)^\top (Y\Omega - XB) \right]. \tag{5}$$

The next theorem establishes the convexity of the above loss as a function of $(B, \Omega)$.

**Theorem 1.** *The negative log-likelihood function $L_n(B, \Omega)$ defined in (5) is convex in $(B, \Omega)$.*

In light of the above theorem, the new parameterization lead to a new convex optimization formulation for multivariate regression. In the next subsection, we demonstrate that by adding appropriate penalty functions, we can still obtain structured estimates for both $C$ and $\Omega$.

As a side note, we would like to point out that another reparameterization $(B, \Theta) = (C\Omega^{1/2}, \Omega^{1/2})$ is more commonly employed in Bayesian statistics for prior specification. It can also be shown that the negative log-likelihood function is convex under this parameterization. However, it is difficult to impose sparsity-inducing on $\Omega$ for this parameterization, which is often desirable when the error covariance is of interest.

## 2.2 Conditional Gaussian Graphical Model

In a conditional Gaussian graphical model [see, e.g., Li et al., 2012, Wang, 2015], the main interest is how to estimate the precision matrix $\Omega$ while adjusting the covariate effect, because the entries in the precision matrix encode the conditional dependence structure among components of the response after adjusting the covariates. A common assumption for the precision matrix is that it is approximately sparse, which is equivalent to saying that conditioned on covariates, every component of the response is only dependent on a small number of other components. Such sparsity assumption can be leveraged by imposing sparsity-inducing penalty on the precision matrix.

More specifically, we use the proposed parameterization in (5) combined with sparsity-inducing penalty on $\Omega$ and group sparsity-inducing penalty on $B$. This allows us to simultaneously obtain a sparse estimate of the precision matrix and a row-sparse estimate of the regression coefficient. In particular, we propose a penalized estimation approach by solving the following optimization problem

$$\underset{B,\Omega}{\text{minimize}} \ \ L_n(B,\Omega) + \lambda_\Omega \sum_{i<j} p_{\tau_\Omega}(\omega_{ij}) + \lambda_B \sum_{i=1}^{p} p_{\tau_B}(\|B_i\|_2), \tag{6}$$

where $\omega_{ij}$ is the $(i,j)$-th entry of $\Omega$, $B_i$ is the $i$-th row of $B$, $(\lambda_\Omega, \lambda_B, \tau_\Omega, \tau_B)$ are tuning parameters, and the truncated lasso penalty function $p_\tau(x) = \min(|x|, \tau)$ [Shen et al., 2013, 2012]. The penalty function is nonconvex, and using a nonconvex penalty is beneficial in several ways. It leads to nearly unbiased parameter estimation, is to facilitate cross-validation for parameter tuning, and can achieve a better sparsity pursuit guarantee under less stringent assumptions [see, e.g., Fan and Li, 2001, Zhang, 2010, Shen et al., 2012, among others]. Also note that other nonconvex penalty such as SCAD and MCP [Fan and Li, 2001, Zhang, 2010] could also be used, and similar theoretical results can be obtained.

One important feature of the proposed parameterization is that the regularization function of $B$ still encourages row sparsity of the original regression coefficient matrix $C$. This is because $B$ and $C$ share the same row-sparsity patterns. Other regularizations are also possible. For example, since $B$ and $C$ share the same rank, we could consider regularization functions that encourage $B$ to have low rank [Yuan et al., 2007]. Due to the space limit, we leave this for future investigations.

The above optimization problem is computationally challenging, especially when the number of responses $q$ is large. The challenge is mainly caused by the $\log \det(\cdot)$ function and the non-smooth penalty functions. A proximal Newton method Lee et al. [2014] can be used to solve this problem. We defer the details into later sections. For a truly large-scale problem, however, we need to consider other alternative loss functions, for which we discuss next.

## 2.3 Two alternative formulations

The $\log \det(\cdot)$ function in the Gaussian log-likelihood function (5) makes the problem computationally prohibitive even for first-order methods. This is because the positive-definiteness needs to be insured for $\Omega$ at every iteration. Here we review two alternative loss functions [Zhang and Zou, 2014, Lee and Hastie, 2015] considered in the Gaussian graphical model literature, and point out that both of them can be adopted into our framework to make the computation more amenable.

More specifically, we consider two loss functions. One is based on a pseudo-likelihood [Lee and Hastie, 2015], and the other one called Dtrace loss is proposed by Zhang and Zou [2014]. In particular, the pseudo-likelihood loss function defined in [Lee and Hastie, 2015] can be expressed as

$$L^{\text{Pseudo}}(\Omega) = \frac{1}{2} \text{Tr} \left[\Omega \Omega_D^{-1} \Omega \Sigma^0\right] - \frac{1}{2} \log \det \Omega_D, \tag{7}$$

and the Dtrace loss function is

$$L^{\text{Dtrace}}(\Omega) = \frac{1}{2}\,\text{Tr}\left[\Omega\Sigma^0\Omega\right] - \text{Tr}(\Omega)\,, \tag{8}$$

where $\Omega_D = \text{Diag}(\Omega)$ denotes a diagonal matrix with diagonal elements equal to that of $\Omega$ and $\Sigma^0 = \left[\Omega^0\right]^{-1}$ is the population covariance matrix. This is in contrast with the negative log-likelihood loss function $L(\Omega) = \frac{1}{2}\,\text{Tr}(\Omega\Sigma^0) - \frac{1}{2}\log\det(\Omega)$, which involves $\log\det(\cdot)$ function. Both loss functions have the desirable properties that

1. they are convex functions of $\Omega$;
2. their gradients vanishes at $\Omega^0$: $\nabla L^{\text{Dtrace}}(\Omega^0) = 0$ and $\nabla L^{\text{Pseudo}}(\Omega^0) = 0$.

This means that these two loss functions can be viewed as Bregman divergences [Brègman, 1966] between $\Omega$ and $\Omega^0$ with certain convex functions. Other alternative loss functions in the Gaussian graphical model literature [see, e.g., Peng et al., 2009, Khare et al., 2015] may also be considered.

In Gaussian graphical model, the empirical loss function can be obtained from substituting $\Sigma^0$ in (7) and (8) with a covariance matrix estimator such as the sample covariance matrix. Similarly, for multivariate regression, we can substitute $\Sigma^0$ with $(Y - XC)^\top (Y - XC)/n$, which leads to the following two empirical losses for multivariate regression

$$L_n^{\text{Pseudo}}(C, \Omega) = \frac{1}{2n}\,\text{Tr}\left[\Omega\Omega_D^{-1}\Omega(Y - XC)^\top(Y - XC)\right] - \frac{1}{2}\log\det\Omega_D\,,$$

$$L_n^{\text{Dtrace}}(C, \Omega) = \frac{1}{2n}\,\text{Tr}\left[\Omega(Y - XC)^\top(Y - XC)\Omega\right] - \text{Tr}(\Omega)\,.$$

Like the negative log-likelihood loss, the above two losses are possibly not convex in $(\Omega, C)$. Fortunately, the following theorem shows that both of them become convex in $(B, \Omega)$ under the proposed new parameterization $(B, \Omega) = (C\Omega, \Omega)$.

**Theorem 2.** *Under the new parameterization $(B, \Omega) = (C\Omega, \Omega)$, the above two loss functions are*

$$L_n^{\text{Pseudo}}(B, \Omega) = \frac{1}{2n}\,\text{Tr}\left[\Omega_D^{-1}(Y\Omega - XB)^\top(Y\Omega - XB)\right] - \frac{1}{2}\log\det\Omega_D\,, \tag{9}$$

$$L_n^{\text{Dtrace}}(B, \Omega) = \frac{1}{2n}\,\text{Tr}\left[(Y\Omega - XB)^\top(Y\Omega - XB)\right] - \text{Tr}(\Omega)\,, \tag{10}$$

*both of which are convex functions of $(B, \Omega)$.*

Consequently, we could replace $L_n(B, \Omega)$ in (6) by $L_n^{\text{Pseudo}}(B, \Omega)$ and $L_n^{\text{Dtrace}}(B, \Omega)$ defined above for faster computation as they no long involve the $\log\det(\cdot)$ function (see the next Section for details).

## 3 Computation

This section discusses optimization methods for solving the proposed penalized multivariate regression problem with the negative log-likelihood loss and the two alternative loss functions. In particular, to treat the nonconvex minimization, we propose to use a difference of convex programming algorithm to solve a sequence of convex relaxed problems. For each relaxed problem, we consider the proximal Newton method by Lee et al. [2014] to solve the one with the negative log-likelihood loss, and a fast first-order method by Beck and Teboulle [2009] to solve the two formulations based on alternative loss functions.

### 3.1 Sequential convex relaxation through DC algorithm

The difference of convex (DC) algorithm is commonly employed for solving nonconvex optimization approximately Shen et al. [2013]. Its key idea is to decompose the objective function into difference of two convex functions, and linearize the trailing function to obtain an upper convex approximation of the nonconvex objective. In our setting, using the DC decomposition that $p_\tau(x) = |x| - \max(|x| - \tau, 0)$, we obtain upper convex approximation of the nonconvex penalty at the previous iterate $(B^{(t)}, \Omega^{(t)})$:

$$p_{\tau_B}\left(\|B_{i\cdot}\|_2\right) \leq \|B_{i\cdot}\|_2 \mathbb{I}(\|B_{i\cdot}^{(t)}\|_2 \leq \tau_B) + \tau_B \mathbb{I}(\|B_{i\cdot}^{(t)}\|_2 > \tau_B)\,,$$

$$p_{\tau_\Omega}\left(|\omega_{ij}|\right) \leq |\omega_{ij}| \mathbb{I}(|\omega_{ij}^{(t)}| \leq \tau_\Omega) + \tau_\Omega \mathbb{I}(|\omega_{ij}^{(t)}| > \tau_\Omega)\,.$$

Accordingly, we solve the nonconvex optimization (6) by considering a sequence of convex relaxations until we get a stationary point. Specifically, we start with some initial estimator $(B^{(0)}, \Omega^{(0)})$ at $t = 0$, to be defined later in Section 4. Then, based on $(B^{(t)}, \Omega^{(t)})$ at step $t$, we consider the following convex relaxation,

$$L_n(B, \Omega) + \lambda_B \sum_{i=1}^{p} \|B_{i\cdot}\|_2 \mathbb{I}(\|B_{i\cdot}^{(t)}\|_2 \leq \tau_B) + \lambda_\Omega \sum_{i<j} |\omega_{ij}| \mathbb{I}(|\omega_{ij}^{(t)}| \leq \tau_\Omega). \tag{11}$$

We then obtain the solution $(B^{(t+1)}, \Omega^{(t+1)})$ at the $(t+1)$th step, and iterate over $t$ until convergence. Typically, only local stationary guarantees is possible. However, we will show later that if the model is well specified and the optimization problem is properly initialized, the stationary solution generated by the DC algorithm coincides with the oracle estimator with probability tending to 1.

## 3.2 Proximal Methods

Standard convex optimization methods can be employed to solve (11). However, they need to be modified to accommodate the non-smoothness. We propose to use the *proximal Newton method* [Lee et al., 2014] for (11) when $L_n(B, \Omega)$ is the negative log-likelihood loss defined in (5). The penalized problem of the two alternative loss functions in (9) and (10) can be solved by first-order proximal method [e.g., Nesterov, 2007, Tseng, 2008, Beck and Teboulle, 2009], where we use the FISTA algorithm by Beck and Teboulle [2009] to solve (11). Details of these algorithm derivations are included in the Appendix.

## 4 Theory

In this section, we derive conditions under which the penalized estimator $\widehat{B}$ and $\widehat{\Omega}$ recovers the oracle estimator $(\widehat{B}^{\mathcal{O}}, \widehat{\Omega}^{\mathcal{O}})$, defined as,

$$(\widehat{B}^{\mathcal{O}}, \widehat{\Omega}^{\mathcal{O}}) = \underset{(\boldsymbol{B}, \boldsymbol{\Omega}): B_i = 0, i \notin S_0, \omega_{ij} = 0, (i,j) \notin A_0}{\arg \min} L_n(B, \Omega), \tag{12}$$

where $A_0 = \{(i, j) : \omega_{ij}^0 \neq 0\}$ and $S_0 = \{i : C_{i\cdot}^0 \neq 0\}$. Note that the oracle estimator is essentially the maximum likelihood estimator given that the sparsity pattern is known. Moreover, rate of convergence for the proposed estimator is also derived.

Since we are dealing with nonconvex penalty, we choose to analyze the stationary point obtained by the DC algorithm with a special initialization scheme. In fact, we plan to show that the DC algorithm stops in one step with the chosen initialization, and recovers the oracle estimator in the second step. More specifically, we obtain an initialized estimator $(\widetilde{B}, \widetilde{\Omega})$ in three steps: (i) first perform $q$ separate Lasso regressions assuming $\Omega = I$ to get an estimator for $C$; (ii) then use the residual error matrix after regressing out covariate effects given estimated regression coefficients $\widetilde{C}$ to get $\widetilde{\Omega}$; (iii) and finally, set $\widetilde{B} = \widetilde{C}\widetilde{\Omega}$.

More specifically, define

$$\widetilde{C}_{\cdot i} = \underset{\beta}{\arg \min} \frac{1}{2n} \cdot \|Y_i - X\beta\|_2^2 + \lambda_B \|\beta\|_1, \tag{13}$$

which is the separate lasso solution. Then, we define a covariance matrix estimate based on the residuals: $\widetilde{\Sigma} = \frac{1}{n}(Y - X\widetilde{C})^\top (Y - X\widetilde{C})$, and let

$$\widetilde{\Omega} = \underset{\sqrt{2a}I \succeq \Omega \succeq 0}{\arg \min} \operatorname{trace}(\widetilde{\Sigma}\Omega) - \log \det \Omega + \sum_{i \neq j} p_{(\lambda_\Omega, a)}(|\omega_{ij}|) \text{ and } \widetilde{B} = \widetilde{C}\widetilde{\Omega}, \tag{14}$$

where $p_{(\lambda, a)}(\cdot)$ is a class of nonconvex penalties defined in Zhu and Li [2018], and $(\lambda, a)$ are tuning parameters. Note that Zhu and Li [2018] shows that (14) is convex for certain range of $\lambda_\Omega$ and $a$, and can be solved efficiently using an alternating direction methods of multipliers (ADMM) algorithm [see, e.g., Boyd et al., 2011].

Now we are ready to present our main results summarized below.

**Theorem 3.** *Under the Assumptions (A1)–(A3) in the Appendix, there exists tuning paramter* $(a, \lambda_B, \lambda_\Omega, \tau_B, \tau_\Omega)$, *such that the estimator obtained from applying the DC algorithm to (6) recovers the oracle estimator with probability at least* $1 - 12\exp(-c\log q)$, *when it is initialized with* $(\widetilde{\Omega}, \widetilde{B})$ *as defined in (14). Here* $c > 0$ *is some absolute constant (see Assumption (A2) in the appendix). Moreover, we have that*

$$\max_{i \in S_0} \frac{\|\widehat{C}^{\mathcal{O}}_{i\cdot} - C^0_{i\cdot}\|_2}{\sqrt{q}} = O_p\left(\sqrt{\frac{\max_i \Sigma^0_{ii}}{\lambda_{\min}\left(\frac{1}{n}X^\top_{S_0}X_{S_0}\right)}\frac{\log(p_0 q)}{n}}\right), \tag{15}$$

$$\|\widehat{\Omega}^{\mathcal{O}} - \Omega^0\|_\infty = O_p\left(\lambda_{\max}(\Sigma^0)\gamma_2\sqrt{\frac{\log q}{n}}\right). \tag{16}$$

The above result says that when the optimization program is properly initialized, the DC algorithm will produce a stationary estimate that recovers the oracle estimator. Moreover, the rate of convergence of the proposed estimator for $\Omega^0$ and $C^0$ in vector $\ell_\infty$ norm scales logarithmically in the ambient dimension ($p$ and $q$). Both rates are optimal up to logarithmic factors.

If we assume that $M, \gamma_1, \gamma_2, \lambda_{\min}(\Sigma^0), \lambda_{\max}(\Sigma^0), \lambda_{\min}\left(\frac{1}{n}X^\top_{S_0}X_{S_0}\right), \phi_0$ are independent of problem dimensions $(n, p, q)$, then the Assumptions (A2)–(A3) can be simplified to

$$n \gtrsim \max(p^2_0, q^2_0)\log(pq)\,, \min_{(i,j)\in A_0}|\omega^0_{ij}| \gtrsim \sqrt{\frac{\log(pq)}{n}}\,, \frac{\min_{i\in S_0}\|C^0_{i\cdot}\|_2}{\sqrt{q}} \gtrsim \max(p_0, q_0)\sqrt{\frac{\log(pq)}{n}}\,.$$

These conditions are commonly imposed assumptions in the literature when analyzing multivariate regression or sparse Gaussian graphical model [see, e.g., Rothman et al., 2010, Liu et al., 2014, Zhu and Li, 2018, and references therein]. Moreover, the rate of convergence results simplify to

$$\|\widehat{\Omega}^{\mathcal{O}} - \Omega^0\|_\infty = O_p\left(\sqrt{\frac{\log q}{n}}\right) \text{ and } \max_{i\in S_0}\frac{\|\widehat{C}^{\mathcal{O}}_{i\cdot} - C^0_{i\cdot}\|_2}{\sqrt{q}} = O_p\left(\sqrt{\frac{\log(p_0 q)}{n}}\right),$$

which are comparable to existing results for Gaussian graphical model [Fan et al., 2009, Shen et al., 2012, Zhu and Li, 2018] when covariates are not present. In this sense, the estimation error of the covariance/precision matrix would not be impacted by the regression coefficient estimation.

## 5    Numerical Studies

This section investigates operating characteristics of the proposed methods with regard to the accuracy of parameter estimation and sparsity recovery of the conditional precision matrix.

In what follows, we consider simulated data from the multivariate regression model $y = C^\top x + \epsilon$ with $\epsilon \sim N(\mathbf{0}, \Omega^{-1})$, where $\Omega$ is the conditional precision matrix. Here the nonzero entries in $\Omega$ encodes an undirected graph, where an edge between two nodes means that the corresponding two variables $y_i$ and $y_j$ are conditionally independent conditioned on covariate $x$. In our numerical experiment, we examine three different types of graphs–a chain graph, a hub graph, and a random graph, as displayed in Figure S1. For a given graph $\mathcal{G} = (\mathcal{V}, \mathcal{E})$, $\mathbf{\Omega}$ is generated based on connectivity of the graph, that is, $\omega_{ij} \neq 0$ if and only if there exists a connection between nodes $i$ and $j$ for $i \neq j$. Moreover, we set $\omega_{ij} = .3$ if $i$ and $j$ are connected and diagonals equal to $.3 + c$ with $c$ chosen so that the smallest eigenvalue of the resulting matrix equals to $.2$. Finally, a random sample is drawn from $N(\mathbf{0}, \mathbf{\Omega}^{-1})$ as the residual in the multivariate regression model. Moreover, we generate the covariate from standard normal distribution, and let coefficient $C$ be a row-sparse matrix, in which the first 3 rows are nonzero and the rest are set to be zero.

With regard to selection of tuning parameters, we fix $\tau_\Omega = .01$ and $\tau_B = .01 \times \sqrt{q}$, and propose to use a vanilla cross-validation to choose the optimal tuning parameter $(\lambda_B, \lambda_\Omega)$ for our methods by minimizing a Kullback-Liebler criterion using a five-fold CV.

We apply the three proposed estimators (denoted as Our-ML, Our-Pseudo, and Our-Dtrace) based on the new parameterizations in Section 2, and compare them to the methods of Rothman et al. [2010] (denoted as MRCE) and Wang [2015] (denoted as Wang) in terms of parameter estimation accuracy and sparse recovery.

We evaluate the accuracy of these method with regard to both parameter estimation and sparse recovery. Specifically, we consider

$$\text{Error}(\widehat{C}) = \text{Tr}\left((\widehat{C} - C^0)\Omega^0(\widehat{C} - C^0)^\top\right), \ \text{Error}(\widehat{\Omega}) = \text{Tr}(\widehat{\Omega}\Sigma^0) - \log\det(\widehat{\Omega}\Sigma^0) - q,$$

$$\text{and Error}(\widehat{C}, \widehat{\Omega}) = \text{Tr}\left((\widehat{C} - C^0)\widehat{\Omega}(\widehat{C} - C^0)^\top\right) + \text{Tr}(\widehat{\Omega}\Sigma^0) - \log\det(\widehat{\Omega}\Sigma^0) - q$$

to measure the estimation error of $\widehat{C}$ alone, $\widehat{\Omega}$ alone, and $(\widehat{C}, \widehat{\Omega})$ jointly. Note that these losses are proportional to the negative log-likelihood loss (up to some constants). For the accuracy of sparse identification, average false positive rate (FPR) and false negative rate (FNR) rates are used:

$$\text{FPR}(\widehat{\Omega}) = \frac{\sum_{1 \le j < j' \le p} \mathbb{I}(\omega_{jj'}^0 = 0, \hat{\omega}_{jj'} \ne 0)}{\sum_{1 \le j < j' \le p} \mathbb{I}(\omega_{jj'}^0 = 0)}\left(1 - \mathbb{I}(\Omega_{\text{off}} \ne 0)\right),$$

$$\text{FNR}(\widehat{\Omega}) = \frac{\sum_{1 \le j < j' \le p} \mathbb{I}(\omega_{jj'}^0 \ne 0, \hat{\omega}_{jj'} = 0)}{\sum_{1 \le j < j' \le p} \mathbb{I}(\omega_{jj'}^0 \ne 0)}\mathbb{I}(\Omega_{\text{off}} \ne 0),$$

$$\text{FPR}(\widehat{C}) = \frac{\sum_{i=1}^p \mathbb{I}(C_{i\cdot}^0 = 0, \widehat{C}_{i\cdot} \ne 0)}{\sum_{i=1}^p \mathbb{I}(C_{i\cdot}^0 = 0)} \ \text{and} \ \text{FNR}(\widehat{C}) = \frac{\sum_{i=1}^p \mathbb{I}(C_{i\cdot}^0 \ne 0, \widehat{C}_{i\cdot} = 0)}{\sum_{i=1}^p \mathbb{I}(C_{i\cdot}^0 \ne 0)}.$$

As shown in Tables S1–S3, the proposed methods outperform both competitors by a large margin in terms of both parameter estimation and sparse recovery across all the situations. In terms of estimation accuracy for the regression coefficients, the amounts of improvement over Wang and MRCE range from 402% to 1159% when the feature dimension is high. Interestingly, the three proposed methods, although based on different loss functions, perform similarly in terms of both estimation and sparse recovery accuracy. This might be due to the high-dimensionality of the problem, as also observed by Zhang and Zou [2014] for the Dtrace loss. With regard to accuracy of sparse recovery, the proposed methods have better balance in terms of minimizing the false positive rate and false negative rate.

Among the three proposed methods, the one based on the negative log-likelihood loss has the highest computational cost. Given that they all perform similarly, the alternative loss functions have an advantage and will be advocated to be used for large-scale problems. That said, the likelihood based method has the advantage that it always produces positive definite estimate for the error covariance matrix, whereas the other two methods sometimes produce infeasible covariance matrix estimate, as indicated by the NAs in Tables S1–S3.

In summary, our simulation results suggest that the proposed methods achieve higher accuracy of sparse identification and parameter estimation, compared to other competitors. We advocate to use the proposed method based on negative log-likelihood loss for small to medium size problems, and to use the methods based on Dtrace and Pseudo loss functions for large-scale problems.

## 6   Discussions

In this article, we proposed a convex optimization formulation for high-dimensional multivariate regression model under three commonly used loss functions. Various optimization algorithms were suggested to solve the associated optimization problems. Our numerical experiments suggest that the estimation of regression coefficients can be improved by incorporating error covariance structure. Moreover, the joint estimation approach leads to more precise sparse identification of the conditional precision matrix compared to existing state-of-art methods. Theoretically, we establish rate of convergence in terms of vector $\ell_\infty$ norm and sparsistency under minimal conditions.

Possible extension to multivariate regression with low rank regularization or constraints could also be considered. It can be shown that the proposed new parameterization preserves the rank of the original parameterization, which makes the extension meaningful. Time series or longitudinal data could be another potential target application of the proposed methodology, where it is of interest to model temporal dependence of the residual after adjusting some covariate effects.

## Broader Impact

This work does not present any foreseeable societal consequence.

## Acknowledgments and Disclosure of Funding

The author is supported in part by the US National Science Foundation (NSF) under grants NSF-DMS-1721445, NSF-DMS-1712580, and NSF-DMS-2015490. The author thanks the reviewers for valuable comments and suggestions that improved the article.

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
