[Supplementary Material]

# Supplementary Material for "A convex optimization formulation for multivariate regression"

This appendix includes the proofs of the technical results in the main article, Figure S1 and Tables S1–S3, and details out the derivations of the proximal Newton and fast proximal gradient algorithm in Section 3 of the main article.

## Proof of Theorem 1

Note that $-\log\det(\Omega)$ is convex [Boyd and Vandenberghe, 2004]. Hence, it suffices to show that $\text{Tr}\left[\Omega^{-1}(Y\Omega - XB)^\top(Y\Omega - XB)\right]$ is convex in $(B, \Omega)$. Toward this end, we write this function as the sum of three functions

$$\text{Tr}\left[\Omega^{-1}(Y\Omega - XB)^\top(Y\Omega - XB)\right] = \text{Tr}(Y^\top Y\Omega) - 2\,\text{Tr}(Y^\top XB) + \text{Tr}(\Omega^{-1}B^\top X^\top XB)$$

The first two parts are linear, and thus are convex. It remains to show the convexity for the last part $\text{Tr}(\Omega^{-1}B^\top X^\top XB)$. We prove this using the fact that $f(x)$ is convex in $x$ if and only if for any $\delta$ and $t \in \mathbb{R}$, $g(t) = f(x + t\delta)$ is convex in $t$. In view of this result, we only need to show that the following univariate function

$$
\begin{aligned}
g(t) &= \text{Tr}\left[(\Omega + tV_1)^{-1}[B + tV_2]^\top X^\top X[B + tV_2]\right] \\
&= \text{Tr}\left[(I + t\Omega^{-1/2}V_1\Omega^{-1/2})^{-1}[B\Omega^{-1/2} + tV_2\Omega^{-1/2}]^\top X^\top X[B\Omega^{-1/2} + tV_2\Omega^{-1/2}]\right]
\end{aligned}
$$

is convex in $t \in \mathbb{R}$ for any $V_1 \in \mathbb{S}^q$ and $V_2 \in \mathbb{R}^{p \times q}$. Since $\Omega^{-1/2} V_1 \Omega^{-1/2}$ is symmetric, we can perform singular value decomposition $\Omega^{-1/2} V_1 \Omega^{-1/2} = UDU^\top$, where $D$ is a diagonal matrix. Substituting this into $g(t)$, it follows that

$$g(t) = \operatorname{Tr}\left[ (I + tD)^{-1} [B\Omega^{-1/2}U + tV_2\Omega^{-1/2}U]^\top X^\top X [B\Omega^{-1/2}U + tV_2\Omega^{-1/2}U] \right]$$
$$= \sum_{i=1}^q \frac{(b_i + tc_i)^\top X^\top X (b_i + tc_i)}{1 + td_i},$$

where $b_i, c_i$ are the $i$-th column of $B\Omega^{-1/2}U$ and $V_2\Omega^{-1/2}U$, respectively. Direct calculation of the second derivative shows that the above function is convex over the domain $1 + td_i > 0, i = 1, \cdots, q$, i.e., $\Omega + tV_1 \succ 0$. This completes the proof.

## Proof of Theorem 2

Note that for the Dtrace loss (10) is a quadratic function with positive semidefinite Hessian, and thence must be convex. For (9), first note that $-\log \det(\Omega_D)$ is convex in $\Omega$. Moreover, the proof of convexity of the first term $\operatorname{Tr}(\Omega_D^{-1}(Y\Omega - XB)^\top (Y\Omega - XB))$ is similar to that of Theorem 1, and is thus omitted. This completes the proof.

## Assumptions for Theorem 3

Throughout, we impose the following technical assumptions for Theorem 3.

**Assumption (A1).** Suppose that a *compatibility condition* is satisfied for the design matrix:

$$\frac{1}{n} \|X\beta\|_2^2 \ge \frac{\phi_0^2}{p_0} \|\beta\|_1^2 \text{ for all } \beta \in \mathbb{R}^p \text{ such that } \|\beta_{S_0^c}\|_1 \le 3\|\beta_{S_0}\|_1 \tag{S1}$$

for some constant $\phi_0 > 0$. Moreover, suppose that

$$\max_{i \in S_0} \|C_{i\cdot}^0\|_2 \le \sqrt{q} M \tag{S2}$$

for some constant $M > 0$.

**Assumption (A2)** (Minimum signal condition)

$$
\min_{(i,j)\in A_0} |\omega_{ij}^0| \;\gtrsim\; (c+2)(\gamma_2 + a(1 + \gamma_1^2\gamma_2))\lambda_{\max}(\Sigma^0)\sqrt{\frac{\log(pq)}{n}}\,, \tag{S3}
$$

$$
\frac{\min_{i\in S_0}\|C_{i\cdot}^0\Omega^0\|_2}{\sqrt{q}} \;\gtrsim\; (c+2)\left(\frac{\sqrt{\kappa(\Sigma^0)}p_0}{\phi_0^2} + M\gamma_2\lambda_{\max}(\Sigma^0)q_0\right)\sqrt{\frac{\log(pq)}{n}}\,, \tag{S4}
$$

where $a \geq \frac{(\lambda_{\max}(\Omega^0)+\gamma_1^{-1}(1+2\gamma_1^2\gamma_2)^{-1})^2}{2}$, $c > 0$ is an absolute constant, and $M, \phi_0 > 0$ refers to constant in (S2) and (S1).

**Assumption (A3)** (Scaling condition)

$$
n \;\gtrsim\; \frac{\gamma_2(c+1)\sqrt{\lambda_{\min}(\Sigma^0)}\max_i \Sigma_{ii}^0}{\phi_0^2}p_0q_0\log(pq)\,, \tag{S5}
$$

$$
n \;\gtrsim\; (c+3)^2\lambda_{\max}^2(\Sigma^0)\gamma_2^2\max\left(\gamma_1^2(1+\gamma_1\gamma_2)^2\,,\,\lambda_{\min}(\Sigma^0)\right)q_0^2\log q\,, \tag{S6}
$$

$$
n \;\gtrsim\; \max\left(1\,,\,\frac{1}{\phi_0^4}\right)p_0^2\log(pq)\,. \tag{S7}
$$

## Proof of Theorem 3

To prove Theorem 3, we first define some notations to be used throughout the proof. Denote by $\Sigma^0$ the covariance matrix of the error, which is a $q \times q$ symmetric positive-definite matrix. Let $\Omega^0 = [\Sigma^0]^{-1} = (\omega_{ij})_{1\leq i,j\leq q}$. Let $A_0$ denote the support of $\Omega^0$, and $q_0 = \max_i \sum_{j=1}^q \omega_{ij}^0 \neq 0$ is the maximum number of nonzeros across rows/columns of $\Omega^0$. Let $\gamma_1 = \|\Sigma^0\|_{\infty,\infty}$ and $\gamma_2 = \|I_{A_0,A_0}^{-1}\|_{\infty,\infty}$, where $I = \nabla^2\left(-\frac{1}{2}\log\det\Omega^0\right)$, $I_{A_0,A_0}$ is the $|A_0|\times|A_0|$ submatrix of $I$ that extracts the corresponding entries whose indices belong to $A_0$, and $|A_0|$ is the size of $A_0$. Let $S_0 = \{i : \|C_{i\cdot}^0\|_2 \neq 0\}$ and $p_0 = |S_0|$. We first present some lemmas to be used later in the proof of the main theorem. The proofs of these lemmas are included in later sections.

**Lemma 1.** *Suppose that rows of $E \in \mathbb{R}^{n\times q}$ are IID $q$-dimensional normal random vectors from*

$N(0, \Sigma^0)$. Let $\widehat{\Sigma}^s = \frac{1}{n} E^\top E$. For any symmetric matrix $T$ and $\nu > 0$

$$\mathbb{P}\left(|\operatorname{Tr}\left((\widehat{\Sigma}^s - \Sigma^0)T\right)| \geq \nu\right) \leq 2\exp\left(-n\frac{\nu^2}{9\|T\|^2 + 8\nu\|T\|}\right), \tag{S8}$$

where $\|T\|^2 = \frac{n}{2}\operatorname{Var}\left(\operatorname{Tr}\left((\widehat{\Sigma}^s - \Sigma^0)T\right)\right)$.

**Lemma 2.** *Suppose that rows of $E \in \mathbb{R}^{n \times q}$ are IID $q$-dimensional normal random vectors from $N(0, \Sigma^0)$. Let*

$$\widehat{\Sigma}^s = \frac{1}{n} E^\top E \;\text{ and }\; \widehat{\Sigma}^e = \frac{1}{n} E^\top (I - P_{X_{S_0}})E,$$

*where $P_{X_{S_0}}$ denotes the projection matrix onto the column space of $X_{S_0}$. Then,*

$$\mathbb{P}\left(\left\|\widehat{\Sigma}^s - \Sigma^0\right\|_\infty \geq \nu \lambda_{\max}(\Sigma^0)\right) \;\leq\; 2\exp\left(-\frac{n\nu^2}{9 + 8\nu} + 2\log q\right),$$

$$\mathbb{P}\left(\left\|\widehat{\Sigma}^e - \Sigma^0\right\|_\infty \geq \left(\nu + \frac{p_0}{n}\right)\lambda_{\max}(\Sigma^0)\right) \;\leq\; 2\exp\left(-\frac{n\nu^2}{9 + 8\nu} + 2\log q\right),$$

*Moreover, if $\max(p_0, \log q) < n$, then for any real anumber $C > 0$, we have that*

$$\mathbb{P}\left(\left\|\widehat{\Sigma}^s - \Sigma^0\right\|_\infty \geq 18(c+2)\lambda_{\max}(\Sigma^0)\sqrt{\frac{\log q}{n}}\right) \leq 2\exp(-c\log q), \tag{S9}$$

$$\mathbb{P}\left(\|\widehat{\Sigma}^e - \Sigma^0\|_\infty \geq \left(\frac{p_0}{n} + 18(c+2)\sqrt{\frac{\log q}{n}}\right)\lambda_{\max}(\Sigma^0)\right) \leq 2\exp(-c\log q). \tag{S10}$$

*As a result,*

$$\left\|\widehat{\Sigma}^s - \Sigma^0\right\|_\infty \;=\; O_p\left(\lambda_{\max}(\Sigma^0)\sqrt{\frac{\log q}{n}}\right),$$

$$\|\widehat{\Sigma}^e - \Sigma^0\|_\infty \;=\; O_p\left(\lambda_{\max}(\Sigma^0)\left(\sqrt{\frac{\log q}{n}} + \frac{p_0}{n}\right)\right).$$

**Lemma 3.** *(Refinement of Theorem 1 of [Zhu and Li [2018]](#)) When*

$$2(1 + \gamma_1^2 \gamma_2)\|\widetilde{\Sigma} - \Sigma^0\|_\infty \leq \lambda_\Omega \leq \frac{\min_{(i,j)\in A_0} |\omega_{ij}^0| - 2\gamma_2\|\widetilde{\Sigma} - \Sigma^0\|_\infty}{a} \tag{S11}$$

$$\text{and } a \geq \frac{(\lambda_{\max}(\Omega^0) + 2\gamma_2 q_0\|\widetilde{\Sigma} - \Sigma^0\|_\infty)^2}{2}, \tag{S12}$$

*problem [(14)](#) is a convex problem, and its minimizer $\widetilde{\Omega}$ satisfies*

$$\|\widetilde{\Omega} - \Omega^0\|_\infty \leq 2\gamma_2\|\widetilde{\Sigma} - \Sigma^0\|_\infty \text{ and } \left\|\widetilde{\Omega}^{-1} - \Sigma^0\right\|_\infty \leq (1 + 2\gamma_1^2\gamma_2)\|\widehat{\Sigma}^e - \Sigma^0\|_\infty \tag{S13}$$

*on the event that*

$$\|\widetilde{\Sigma} - \Sigma^0\|_\infty \leq \min\left\{\frac{\min_{(i,j)\in A_0} |\omega_{ij}^0|}{2\gamma_2 + 2a(1 + \gamma_1^2\gamma_2)}, \frac{\sqrt{2a} - \lambda_{\max}(\Omega^0)}{2\gamma_2 q_0}, \frac{1}{2\gamma_1\gamma_2(1 + 2\gamma_1^2\gamma_2)q_0}\right\}, \tag{S14}$$

**Lemma 4.** *(Convergence rate of the initializer $(\widetilde{B}, \widetilde{\Omega})$) Let*

$$\begin{aligned}
\mathcal{J} &= \cap_{i=1}^q \left\{ \frac{2}{n}\|X^\top E_{\cdot i}\|_\infty \leq \sqrt{2\Sigma_{ii}^0 \frac{(c+1)\log q + \log p}{n}} \right\}, \\
\mathcal{K} &= \left\{ \|\widehat{\Sigma}^s - \Sigma^0\|_\infty \leq \frac{1}{4}\min\left(\frac{\min_{(i,j)\in A_0} |\omega_{ij}^0|}{\gamma_2 + a(1 + \gamma_1^2\gamma_2)}, \frac{1}{\gamma_1\gamma_2(1 + 2\gamma_1^2\gamma_2)q_0}\right) \right\},
\end{aligned} \tag{S15}$$

*where $a \geq \frac{(\lambda_{\max}(\Omega^0) + \gamma_1^{-1}(1 + 2\gamma_1^2\gamma_2)^{-1})^2}{2}$. Then, under [(S5)](#) in Assumption (A1), and on the event that $\mathcal{K} \cap \mathcal{J}$, the initial estimators $(\widetilde{B}, \widetilde{\Omega})$ defined above with*

$$\lambda_B = 2\sqrt{\frac{2(c+1)\max_i \Sigma_{ii}^0 \log(pq)}{n}}, \tag{S16}$$

$$2(1 + \gamma_1^2\gamma_2)\left(\|\widehat{\Sigma}^s - \Sigma^0\|_\infty + \frac{48(c+1)\max_i \Sigma_{ii}^0}{\phi_0^2} \cdot \frac{p_0 \log(pq)}{n}\right) \leq \lambda_\Omega$$

$$\leq \frac{\min_{(i,j)\in A_0} |\omega_{ij}^0| - 2\gamma_2\left(\|\widehat{\Sigma}^s - \Sigma^0\|_\infty + \frac{48(c+1)\max_i \Sigma_{ii}^0}{\phi_0^2} \cdot \frac{p_0 \log(pq)}{n}\right)}{a}, \tag{S17}$$

$$a \geq \frac{(\lambda_{\max}(\Omega^0) + \gamma_1^{-1}(1 + 2\gamma_1^2\gamma_2)^{-1})^2}{2} \tag{S18}$$

*satisfies the following bound*

$$\|\widetilde{\Omega} - \Omega^0\|_\infty \;\leq\; 2\gamma_2 \|\widehat{\Sigma}^s - \Sigma^0\|_\infty + \frac{96(c+1)\gamma_2 \max_i \Sigma_{ii}^0}{\phi_0^2} \cdot \frac{p_0 \log(pq)}{n}\,, \tag{S19}$$

$$\frac{\|\widetilde{B}_{i\cdot} - B_{i\cdot}^0\|_2}{\sqrt{q}} \;\leq\; \frac{16\sqrt{(c+1)\kappa(\Sigma^0)}}{\phi_0^2}\sqrt{\frac{p_0^2 \log(pq)}{n}} + q_0 \|\widetilde{\Omega} - \Omega^0\|_\infty \frac{\|C_{i\cdot}^0\|_2}{\sqrt{q}} \tag{S20}$$

*for $i = 1, \ldots p$. Moreover, we have that*

$$\mathbb{P}(\mathcal{J}) \geq 1 - 2\exp(-c\log q)\,, \tag{S21}$$

*and under (S6) in Assumption (A2), we have that*

$$\mathbb{P}(\mathcal{K}) \geq 1 - 2\exp(-c\log q)\,. \tag{S22}$$

The next lemma derives the rate of convergence for the oracle estimator.

**Lemma 5.** *(Existence, uniqueness and properties of the oracle estimator) On the event that $\mathcal{G} \cap \mathcal{W}$ with*

$$\mathcal{G} \;=\; \left\{ \|\widehat{\Sigma}^e - \Sigma^0\|_\infty \leq \min\left( \frac{1}{\gamma_1\gamma_2(1 + 2\gamma_1^2\gamma_2)q_0}, \frac{\sqrt{\lambda_{\max}(\Omega^0)}}{\gamma_2 q_0} \right) \right\},$$

$$\mathcal{W} \;=\; \left\{ \frac{1}{n}\left\| \left(\frac{1}{n}X_{S_0}^\top X_{S_0}\right)^{-1} X_{S_0}^\top E \right\|_{\infty,2} \leq C\lambda_{\min}^{-1}\left\{ \left(\frac{1}{n}X_{S_0}^\top X_{S_0}\right) \right\} \sqrt{\max_i \Sigma_{ii}^0 \frac{q\log(pq)}{n}} \right\}$$

$$\cap \left\{ \|X^\top(I - P_{X_{S_0}})E\|_{\infty,2} \leq C\sqrt{\max_i \Sigma_{ii}^0 \frac{q\log(pq)}{n}} \right\},$$

*the oracle estimator* $(\widehat{B}^{\mathcal{O}}, \widehat{\Omega}^{\mathcal{O}})$ *exists and is unique, and we have that*

$$\|\widehat{\Omega}^{\mathcal{O}} - \Omega^0\|_\infty \leq 2\gamma_2\|\widehat{\Sigma}^e - \Sigma^0\|_\infty \,, \tag{S23}$$

$$\|\nabla_\Omega L_n(\widehat{\Omega}^{\mathcal{O}}, \widehat{B}^{\mathcal{O}})\|_\infty \leq 2(1 + \gamma_1^2\gamma_2)\|\widehat{\Sigma}^e - \Sigma^0\|_\infty \,, \tag{S24}$$

$$\frac{\|\widehat{B}_{i\cdot}^{\mathcal{O}} - B_{i\cdot}^0\|_2}{\sqrt{q}} \leq 3\sqrt{\frac{(c+1)\kappa(\Sigma^0)}{\lambda_{\min}\left(\frac{1}{n}X_{S_0}^\top X_{S_0}\right)}\frac{\log(p_0 q)}{n}}$$

$$+ q_0\|\widehat{\Omega}^{\mathcal{O}} - \Omega^0\|_\infty \frac{\|C_{i\cdot}^0\|_2}{\sqrt{q}} \text{ for all } i \in S_0 \,, \tag{S25}$$

$$\frac{\max_i \left\|\nabla_{B_{i\cdot}} L_n(\widehat{\Omega}^{\mathcal{O}}, \widehat{B}^{\mathcal{O}})\right\|_2}{\sqrt{q}} \leq 2\sqrt{(c+1)\max_i \Sigma_{ii}^0 \frac{\log(pq)}{n}} \,. \tag{S26}$$

*Moreover, under the scaling condition* (S6) *and* (S7) *in Assumption (A3), we have that*

$$\mathbb{P}(\mathcal{W}) \geq 1 - 6\exp\left(-c\log(p_0 q)\right) \,, \tag{S27}$$

$$\mathbb{P}(\mathcal{G}) \geq 1 - 2\exp(-c\log q) \,. \tag{S28}$$

Now we are ready to prove Theorem 3. We prove the claim in two steps. In the first step, we show that when initialized by $(\widetilde{B}, \widetilde{\Omega})$, the very first iterate of the DC algorithm coincides with the oracle estimator $(\widehat{B}^{\mathcal{O}}, \widehat{\Omega}^{\mathcal{O}})$. In the second step, we show that the DC algorithm stops at the second iterate, and recovers the oracle estimator. Throughout, all the statement are stated under the event: $\mathcal{J} \cap \mathcal{K} \cap \mathcal{G} \cap \mathcal{W}$, which by using Lemma 4 and 5, has probability at least

$$\mathbb{P}(\mathcal{J} \cap \mathcal{K} \cap \mathcal{G} \cap \mathcal{W}) \geq 1 - 12\exp(-c\log q) \,.$$

For the first step, it suffices to prove that there exist $(\tau_B, \tau_\Omega)$ and $(\lambda_B, \lambda_\Omega)$ such that

$$\min_{i \in S_0} \|\widetilde{B}_{i\cdot}\|_2 \geq \tau_B \geq \max_{i \notin S_0} \|\widetilde{B}_{i\cdot}\|_2 = \max_{i \notin S_0} \|\widetilde{B}_{i\cdot} - B_{i\cdot}^0\|_2 \,, \tag{S29}$$

$$\min_{(i,j) \in A_0} |\widetilde{\omega}_{ij}| \geq \tau_\Omega \geq \max_{(i,j) \notin A_0} |\widetilde{\omega}_{ij}| = \max_{(i,j) \notin A_0} |\widetilde{\omega}_{ij} - \omega_{ij}^0| \,, \tag{S30}$$

and

$$\left\|\nabla_{B_i} L_n(\widehat{B}^{\mathcal{O}}, \widehat{\Omega}^{\mathcal{O}})\right\|_2 \le \lambda_B \text{ for } i \notin S_0 \text{ and } \left|\frac{\partial L_n(\widehat{B}^{\mathcal{O}}, \widehat{\Omega}^{\mathcal{O}})}{\partial \omega_{ij}}\right| \le \lambda_\Omega \text{ for } (i,j) \notin A_0 \,, \qquad \text{(S31)}$$

$$\nabla_{B_i} L_n(\widehat{B}^{\mathcal{O}}, \widehat{\Omega}^{\mathcal{O}}) = 0 \text{ for } i \in S_0 \text{ and } \frac{\partial L_n(\widehat{B}^{\mathcal{O}}, \widehat{\Omega}^{\mathcal{O}})}{\partial \omega_{ij}} = 0 \text{ for } (i,j) \in A_0 \,. \qquad \text{(S32)}$$

This is because under (S29) and (S30), the relaxed convex objective at $(\tilde{B}, \tilde{\Omega})$ (i.e., (11)) becomes

$$L_n(B, \Omega) + \lambda_B \sum_{i \notin S_0} \|B_{i\cdot}\|_2 + \lambda_\Omega \sum_{(i,j) \notin A_0} |\omega_{ij}| \,, \qquad \text{(S33)}$$

and the optimality conditions (S31) and (S32) ensures that $(\widehat{B}^{\mathcal{O}}, \widehat{\Omega}^{\mathcal{O}})$ is a minimizer of (S33).

To prove the existence of $(\tau_B, \tau_\Omega)$ and $(\lambda_B, \lambda_\Omega)$ that satisfies conditions (S29)–(S32), we first note that

$$\min_{i \in S_0} \|\widetilde{B}_{i\cdot}\|_2 \ge \min_{i \in S_0} \left\{\|B_{i\cdot}^0\|_2 - \|\widetilde{B}_{i\cdot} - B_{i\cdot}^0\|_2\right\} \ge \min_{i \in S_0} \|B_{i\cdot}^0\|_2 - \max_{i \in S_0} \|\widetilde{B}_{i\cdot} - B_{i\cdot}^0\|_2 \,,$$

$$\min_{(i,j) \in A_0} |\widetilde{\omega}_{ij}| \ge \min_{(i,j) \in A_0} \left\{|\omega_{ij}^0| - |\widetilde{\omega}_{ij} - \omega_{ij}^0|\right\} \ge \min_{(i,j) \in A_0} |\omega_{ij}^0| - \max_{(i,j) \in A_0} |\widetilde{\omega}_{ij} - \omega_{ij}^0| \,.$$

Then, applying Lemma 2 and 4, we know that under Assumption (A1) and (A2), we have that

$$\|\widetilde{B}_{i\cdot} - B_{i\cdot}^0\|_2 \le \frac{16\sqrt{(c+1)\kappa(\Sigma^0)}}{\phi_0^2} \sqrt{\frac{p_0^2 q \log(pq)}{n}} \text{ for } i \notin S_0 \qquad \text{(S34)}$$

and for $i \in S_0$,

$$
\begin{aligned}
\|\widetilde{B}_{i\cdot} - B_{i\cdot}^0\|_2 &\leq \frac{16\sqrt{(c+1)\kappa(\Sigma^0)}}{\phi_0^2}\sqrt{\frac{p_0^2 q \log(pq)}{n}} + q_0\|\widetilde{\Omega} - \Omega^0\|_\infty \max_i \|C_{i\cdot}^0\|_2 \\
&\leq \frac{16\sqrt{(c+1)\kappa(\Sigma^0)}}{\phi_0^2}\sqrt{\frac{p_0^2 q \log(pq)}{n}} + \\
&\quad q_0\left(2\gamma_2\|\widehat{\Sigma}^s - \Sigma^0\|_\infty + \frac{96(c+1)\gamma_2 \max_i \Sigma_{ii}^0}{\phi_0^2}\frac{p_0 \log(pq)}{n}\right) \max_i \|C_{i\cdot}^0\|_2 \\
&\leq \frac{16\sqrt{(c+1)\kappa(\Sigma^0)}}{\phi_0^2}\sqrt{\frac{p_0^2 q \log(pq)}{n}} + \\
&\quad 2\gamma_2 q_0\left(18(c+2)\lambda_{\max}(\Sigma^0)\sqrt{\frac{\log q}{n}} + \frac{48(c+1)\max_i \Sigma_{ii}^0}{\phi_0^2}\frac{p_0 \log(pq)}{n}\right) \max_i \|C_{i\cdot}^0\|_2 \\
&\leq \frac{16\sqrt{(c+1)\kappa(\Sigma^0)}}{\phi_0^2}\sqrt{\frac{p_0^2 q \log(pq)}{n}} + 38\gamma_2 q_0(c+2)\lambda_{\max}(\Sigma^0)\sqrt{\frac{\log(pq)}{n}} \max_i \|C_{i\cdot}^0\|_2
\end{aligned}
$$

provided that
$$
\frac{48(c+1)\max_i \Sigma_{ii}^0}{\phi_0^2}\frac{p_0 \log(pq)}{n} \leq (c+2)\lambda_{\max}(\Sigma^0)\sqrt{\frac{\log(pq)}{n}},
$$

which is ensured by the scaling condition (S7) in Assumption (A3). Hence, a sufficient condition for (S29) is

$$
\begin{aligned}
\min_{i\in S_0}\|B_{i\cdot}^0\|_2 - \frac{16\sqrt{(c+1)\kappa(\Sigma^0)}}{\phi_0^2}\sqrt{\frac{p_0^2 q \log(pq)}{n}} &- 38\gamma_2(c+2)\lambda_{\max}(\Sigma^0)\sqrt{\frac{q_0^2 \log(pq)}{n}} \max_i \|C_{i\cdot}^0\|_2 \\
&\geq \tau_B \geq \frac{16\sqrt{(c+1)\kappa(\Sigma^0)}}{\phi_0^2}\sqrt{\frac{p_0^2 q \log(pq)}{n}}.
\end{aligned} \tag{S35}
$$

The existence of $\tau_B$ satisfying the above condition is ensured by (S4).

Similarly, by applying Lemma 2 and 4, we know that under Assumption (A1) and (A2), we have

that

$$\max_{(i,j)} |\widetilde{\omega}_{ij} - \omega_{ij}^0|$$

$$\leq 2\gamma_2 \|\widehat{\Sigma}^s - \Sigma^0\|_\infty + \frac{96(c+1)\gamma_2 \max_i \Sigma_{ii}^0}{\phi_0^2} \cdot \frac{p_0 \log(pq)}{n}$$

$$\leq 36\gamma_2(c+2)\lambda_{\max}(\Sigma^0)\sqrt{\frac{\log q}{n}} + \frac{96(c+1)\gamma_2 \max_i \Sigma_{ii}^0}{\phi_0^2} \cdot \frac{p_0 \log(pq)}{n}$$

$$\leq 38\gamma_2(c+2)\lambda_{\max}(\Sigma^0)\sqrt{\frac{\log(pq)}{n}}$$

provided that

$$\frac{96(c+1)\gamma_2 \max_i \Sigma_{ii}^0}{\phi_0^2} \cdot \frac{p_0 \log(pq)}{n} \leq 2\gamma_2(c+2)\lambda_{\max}(\Sigma^0)\sqrt{\frac{\log(pq)}{n}}$$

which is ensured by the scaling condition (S7) in Assumption (A3). Hence, a sufficient condition for (S30) is

$$\min_{(i,j)\in A_0} |\omega_{ij}^0| - 38\gamma_2(c+2)\lambda_{\max}(\Sigma^0)\sqrt{\frac{\log(pq)}{n}} \geq \tau_\Omega \geq 38\gamma_2(c+2)\lambda_{\max}(\Sigma^0)\sqrt{\frac{\log(pq)}{n}} \qquad \text{(S36)}$$

Moreover, by using Lemma 2 and 5, we have that

$$\max_{i\notin S_0} \left\| \nabla_{B_{i\cdot}} L_n(\widehat{B}^{\mathcal{O}}, \widehat{\Omega}^{\mathcal{O}}) \right\|_2 \leq 2\sqrt{(c+1)\max_i \Sigma_{ii}^0 \frac{\log(pq)}{n}}, \qquad \text{(S37)}$$

and

$$\max_{(i,j)\notin A_0} \left| \frac{\partial L_n(\widehat{B}^{\mathcal{O}}, \widehat{\Omega}^{\mathcal{O}})}{\partial \omega_{ij}} \right| \leq 2(1+\gamma_1^2\gamma_2)\|\widehat{\Sigma}^e - \Sigma^0\|_\infty$$

$$\leq 2(1+\gamma_1^2\gamma_2)\lambda_{\max}(\Sigma^0)\left( \frac{p_0}{n} + 18(c+2)\sqrt{\frac{\log q}{n}} \right)$$

$$\leq 38(c+2)(1+\gamma_1^2\gamma_2)\lambda_{\max}(\Sigma^0)\sqrt{\frac{\log q}{n}}$$

Hence, if $\lambda_B$ and $\lambda_\Omega$ satisfy

$$\lambda_\Omega \geq 38(c+2)(1+\gamma_1^2\gamma_2)\lambda_{\max}(\Sigma^0)\sqrt{\frac{\log q}{n}} \text{ and } \lambda_B \geq 2\sqrt{(c+1)\max_i \Sigma_{ii}^0 \frac{\log(pq)}{n}}, \qquad \text{(S38)}$$

then (S31) holds. Note that the above conditions are ensured by (S16) and (S17), under the event $\mathcal{J} \cap \mathcal{K}$. Lastly, by definition of the oracle estimator, (S32) holds since they are the score equation of the oracle estimator. This completes the proof of the first step.

For the second step, it suffices to show that the oracle estimator satisfies

$$\|\widehat{B}_i^{\mathcal{O}}\|_2 \geq \tau_B \text{ for } i \in S_0 \text{ and } |\widehat{\omega}_{ij}^{\mathcal{O}}| \geq \tau_\Omega \text{ for } (i,j) \in A_0. \qquad \text{(S39)}$$

for any $\tau_B$ and $\tau_\Omega$ satisfying (S35) and (S36) (which also ensures (S29) and (S30)).

Again, this is because (S39) implies that the relaxed convex problem at $(\widehat{B}^{\mathcal{O}}, \widehat{\Omega}^{\mathcal{O}})$ (i.e., (11)) is

$$L_n(B,\Omega) + \lambda_B \sum_{i \notin S_0} \|B_{i\cdot}\|_2 + \lambda_\Omega \sum_{(i,j) \notin A_0} |\omega_{ij}|,$$

and the oracle estimator $(\widehat{B}^{\mathcal{O}}, \widehat{\Omega}^{\mathcal{O}})$ minimizes this function.

Now, applying Lemma 2 and 5, we have that

$$\min_{i \in S_0} \|\widehat{B}_i^{\mathcal{O}}\|_2 \geq \min_{i \in S_0} \|B_{i\cdot}^0\|_2 - \max_{i \in S_0} \|\widehat{B}_i^{\mathcal{O}} - B_{i\cdot}^0\|_2 \qquad \text{(S40)}$$

$$\geq \min_{i \in S_0} \|B_{i\cdot}^0\|_2 - 3\sqrt{\frac{(c+1)\kappa(\Sigma^0)}{\lambda_{\min}\left(\frac{1}{n}X_{S_0}^\top X_{S_0}\right)} \frac{q\log(p_0q)}{n}} - q_0\|\widehat{\Omega}^{\mathcal{O}} - \Omega^0\|_\infty \max_i \|C_{i\cdot}^0\|_2 \qquad \text{(S41)}$$

Note that

$$\|\widehat{\Omega}^{\mathcal{O}} - \Omega^0\|_\infty \leq 2\gamma_2\|\widehat{\Sigma}^e - \Sigma^0\|_\infty \leq 2\gamma_2\lambda_{\max}(\Sigma^0)\left(\frac{p_0}{n} + 18(c+2)\sqrt{\frac{\log q}{n}}\right)$$

$$\leq 38\gamma_2\lambda_{\max}(\Sigma^0)(c+2)\sqrt{\frac{\log q}{n}} \qquad \text{(S42)}$$

under the scaling condition (S7). Hence, a sufficient condition for the first inequality in (S39) would be

$$
\begin{aligned}
\tau_B \;\leq\; & \min_{i \in S_0} \|B_{i\cdot}^0\|_2 - 3\sqrt{\frac{(c+1)\kappa(\Sigma^0)}{\lambda_{\min}\left(\frac{1}{n}X_{S_0}^\top X_{S_0}\right)}\frac{q\log(p_0 q)}{n}} \\
& -38\gamma_2 \lambda_{\max}(\Sigma^0)(c+2)\sqrt{\frac{q_0^2 \log q}{n}}\, \max_i \|C_{i\cdot}^0\|_2\,,
\end{aligned}
\tag{S43}
$$

which is ensured by (S35), because, by (S50) and the fact that

$$
\lambda_{\min}\left(\frac{1}{n}X_{S_0}^\top X_{S_0}\right) \leq \min_{i \in S_0} n^{-1}\|X_{\cdot i}\|_2^2 = 1 \text{ and}
$$
$$
p_0/\phi_0^2 \geq \lambda_{\min}^{-1}\left(\frac{1}{n}X_{S_0}^\top X_{S_0}\right) \geq \lambda_{\min}^{-1/2}\left(\frac{1}{n}X_{S_0}^\top X_{S_0}\right)\,.
$$

Hence, the existence of $\tau_B$ can be ensured by (S4), because $\max_i \|C_{i\cdot}^0\|_2 \leq \sqrt{q}M$ by assumption. Similarly,

$$
\begin{aligned}
& \min_{(i,j)\in A_0} |\widehat{\omega}_{ij}^{\mathcal{O}}| \\
\geq\; & \min_{(i,j)\in A_0} |\omega_{ij}^0| - \|\widehat{\Omega}^{\mathcal{O}} - \Omega^0\|_\infty \geq \min_{(i,j)\in A_0} |\omega_{ij}^0| - 2\gamma_2 \|\widehat{\Sigma}^e - \Sigma^0\|_\infty \\
\geq\; & \min_{(i,j)\in A_0} |\omega_{ij}^0| - 2\lambda_{\max}(\Sigma^0)\gamma_2\left(\frac{p_0}{n} + 18(c+2)\sqrt{\frac{\log q}{n}}\right) \\
\geq\; & \min_{(i,j)\in A_0} |\omega_{ij}^0| - 38\lambda_{\max}(\Sigma^0)\gamma_2(c+2)\sqrt{\frac{\log q}{n}}
\end{aligned}
$$

Hence, a sufficient condition for the second inequality in (S39) would be

$$
\tau_\Omega \leq \min_{(i,j)\in A_0} |\omega_{ij}^0| - 38\lambda_{\max}(\Sigma^0)\gamma_2(c+2)\sqrt{\frac{\log q}{n}}\,,
$$

which is ensured by (S36). Hence, the existence of $\tau_\Omega$ is ensured by (S3). The oracle bounds (16) and (15) follow immediately from (S42) and (S63). This compeletes the proof.

# Proofs of Lemmas 1-5

**Proof of Lemma 1:** By Markov's inequality, for any $\nu > 0$,

$$
\begin{aligned}
P\left(\operatorname{Tr}\left((\widehat{\Sigma}^s - \Sigma^0)T\right) \geq \nu\right) &\leq \exp\left(-\frac{\gamma\sqrt{n}\nu}{2}\right) \mathbb{E}\exp\left(\frac{\gamma\sqrt{n}}{2}\operatorname{Tr}\left((\widehat{\Sigma}^s - \Sigma^0)T\right)\right) \\
&\leq \exp\left(\underbrace{\log\mathbb{E}\exp\left(\frac{\gamma\sqrt{n}}{2}\operatorname{Tr}\left((\widehat{\Sigma}^s - \Sigma^0)T\right)\right) - \frac{\gamma\sqrt{n}\nu}{2}}_{I_1}\right),
\end{aligned}
$$

where $\gamma$ is chosen such that $\gamma \in \left[0, \frac{M_0\sqrt{n}}{\|\sqrt{\Sigma^0}T\sqrt{\Sigma^0}\|_F}\right]$ for some constant $0 < M_0 < 1$, which is to be determined later. Moreover, after some calculations, we have that

$$
\begin{aligned}
\mathbb{E}\exp\left(\frac{\gamma\sqrt{n}}{2}\operatorname{Tr}\left((\widehat{\Sigma}^s - \Sigma^0)T\right)\right) &= \left(\mathbb{E}\exp\left(\frac{\gamma\sqrt{n}}{2}\operatorname{Tr}\left((ee^T - \Sigma^0)T\right)\right)\right)^n \\
&= \exp\left(-\frac{\gamma\sqrt{n}}{2}\operatorname{Tr}(\Sigma^0 T)\right)\det\left(\boldsymbol{I} - \frac{\gamma}{\sqrt{n}}\Sigma^0 T\right)^{-n/2}, \quad \text{(S44)}
\end{aligned}
$$

where $e \sim N(0, \Sigma^0)$ and the last equality requires that $\sqrt{n}\Omega^0 \succeq \gamma T$, which is ensured by the fact that $\gamma \leq \frac{M_0\sqrt{n}}{\|\sqrt{\Sigma^0}T\sqrt{\Sigma^0}\|_F} < \frac{\sqrt{n}}{\|\sqrt{\Sigma^0}T\sqrt{\Sigma^0}\|_F}$. Consequently,

$$
\log\mathbb{E}\exp\left(\frac{\gamma\sqrt{n}}{2}\operatorname{Tr}\left((\widehat{\Sigma}^s - \Sigma^0)T\right)\right) = \log\det\left(I - \frac{\gamma}{\sqrt{n}}\Sigma^0 T\right)^{-n/2} - \frac{\gamma\sqrt{n}}{2}\operatorname{Tr}(\Sigma^0 T). \quad \text{(S45)}
$$

An expansion of the log det function gives

$$
\begin{aligned}
&\log\det(I - \frac{\gamma}{\sqrt{n}}\Sigma^0 T)^{-n/2} \\
&= \frac{\gamma\sqrt{n}}{2}\operatorname{Tr}(\Sigma^0 T) + \frac{\gamma^2}{4}\operatorname{Tr}((\Sigma^0 T)^2) + \underbrace{\frac{n}{2}\sum_{l=3}^{\infty}l^{-1}\operatorname{Tr}\left((\frac{\gamma\Sigma^0 T}{\sqrt{n}})^l\right)}_{I_2}. \quad \text{(S46)}
\end{aligned}
$$

For $I_2$, note that $I_2 \leq \frac{n}{2}\sum_{l=3}^{\infty}l^{-1}\left(\frac{\gamma\|T\|}{\sqrt{n}}\right)^l \leq \gamma^2\|T\|^2\frac{3-M_0}{12(1-M_0)}$. Similarly, $I_1 \leq \frac{M_1+1}{4}\gamma^2\|T\|^2 - \frac{\gamma\sqrt{n}\nu}{2}$, where $M_1 = \frac{3-M_0}{3(1-M_0)}$. Minimizing this upper bound of $I_1$ as a function of $\gamma$ over the interval

$\left[0, \frac{M_0\sqrt{n}}{\|T\|}\right]$, we obtain that

$$
\begin{aligned}
I_1 &\le -\frac{n\nu^2}{4(1+M_1)\|T\|^2} && \text{if } \nu \le M_0(1+M_1)\|T\| \\
I_1 &\le -\frac{nM_0}{2\|T\|}\left(\nu - \frac{M_0(1+M_1)}{2}\|T\|\right) && \text{otherwise.}
\end{aligned}
$$

A combination of these two cases yields that $I_1 \le -\frac{nM_0\nu^2}{4M_0(M_1+1)\|T\|^2+2\nu\|T\|}$. Set $M_0 = 4^{-1}$, and then $M_1 = 11/9$, we obtain the desired results

$$
P\left(\operatorname{Tr}\left((\widehat{\Sigma}^s - \Sigma^0)T\right) \ge \nu\right) \le \exp\left(-n\frac{\nu^2}{9\|T\|^2 + 8\nu\|T\|}\right),
$$

for any $\nu > 0$. The other direction follows exactly the same argument, and thus is omitted. This completes the proof.

**Proof of Lemma 2.** Applying Lemma 1 with $T = T_k; k = 1, \ldots, T_{q^2}$ with $\{T_1, \cdots, T_{q^2}\} = \left\{(e_i^\top e_j + e_j^\top e_i)/2\right\}_{1\le i,j\le q}$ then applying an inequality $\|\sqrt{\Sigma^0}(e_i^\top e_j + e_j^\top e_i)\sqrt{\Sigma^0}/2\|_F^2 \le \lambda_{\max}(\Sigma^0)$ and a union bound, we obtain that

$$
\mathbb{P}\left(\left\|\widehat{\Sigma}^s - \Sigma^0\right\|_\infty \ge \nu\lambda_{\max}(\Sigma^0)\right) \le 2\exp\left(-n\frac{\nu^2}{9+8\nu} + 2\log q\right) \tag{S47}
$$

Now by (S47), for any $R > 1$ and $\frac{\log q}{n} < 1$, we have that

$$
\begin{aligned}
\mathbb{P}\left(\left\|\widehat{\Sigma}^s - \Sigma^0\right\|_\infty \ge R\lambda_{\max}(\Sigma^0)\sqrt{\frac{\log q}{n}}\right) &\le 2\exp\left(-\frac{R^2\log q}{9+8R\sqrt{\frac{\log q}{n}}} + 2\log q\right) \\
&\le 2\exp\left\{-\log(q)\left(\frac{R^2}{9+8R\sqrt{\frac{\log q}{n}}} - 2\right)\right\} \\
&\le 2\exp\left(-((R/18) - 2)\log q\right). \tag{S48}
\end{aligned}
$$

Hence, (S9) follows from setting $R = 18(c+2)$. Hence, when $\frac{\log q}{n} = O(1)$, we have that

$$\left\| \widehat{\Sigma}^s - \Sigma^0 \right\|_\infty = O_p\left( \lambda_{\max}(\Sigma^0) \sqrt{\frac{\log q}{n}} \right)$$

Next, we bound $\|\widehat{\Sigma}^e - \Sigma^0\|_\infty$. Note that

$$
\begin{aligned}
\widehat{\Sigma}^e - \Sigma^0 &= (1/n) \cdot Y^\top (I - P_{X_{S_0}}) Y \\
&= (1/n) \cdot (X_{S_0} C^0_{S_0} + E)^\top (I - P_{X_{S_0}})(X_{S_0} C^0_{S_0} + E) - \Sigma^0 \\
&= (1/n) \cdot E^\top (I - P_{X_{S_0}}) E - \Sigma^0 = (1/n) \cdot E^\top U^\top D U E - \Sigma^0,
\end{aligned}
$$

where we have performed singular value decomposition $I - P_{X_{S_0}} = U^\top D U$. It is easy to verify that the rows of $UE$ are still independent and identically distributed normal random variables with mean 0 and covariance matrix $\Sigma^0$, because

$$\mathrm{cov}(u_i^\top e_k, u_j^\top e_l) = \sum_{m=1}^n u_{im} u_{jm} \mathrm{cov}(e_{mk}, e_{ml}) = \sigma^0_{kl} u_i^\top u_j = \sigma^0_{kl} \mathbb{I}(i = j),$$

where $u_i$ and $e_k$ are the $i$-th row and $k$-th column of $U$ and $E$, respectively. Since $I - P_{X_{S_0}}$ is idempotent with rank $\min(n, p_0)$, we have that $E^\top U^\top D U E = \tilde{E}^\top \tilde{E}$, where $E$ is a $(n - \min(p_0, n)) \times q$ random matrix where its rows are IID from $N(0, \Sigma^0)$.

Now, note that

$$
\begin{aligned}
\|\widehat{\Sigma}^e - \Sigma^0\|_\infty &= \left\| \frac{1}{n} \tilde{E}\tilde{E} - \Sigma^0 \right\|_\infty \leq \frac{n - p_0}{n} \left\| \frac{1}{n - p_0} \tilde{E}\tilde{E} - \Sigma^0 \right\|_\infty + \frac{p_0}{n} \|\Sigma^0\|_\infty \\
&\leq \frac{n - p_0}{n} \left\| \frac{1}{n - p_0} \tilde{E}\tilde{E} - \Sigma^0 \right\|_\infty + \frac{p_0}{n} \lambda_{\max}(\Sigma^0)
\end{aligned}
$$

where we have used the fact that $\|\Sigma^0\|_\infty \leq \lambda_{\max}(\Sigma^0)$, and $p_0 \leq n$.

Again, by applying (S47), we have that

$$
\begin{aligned}
\mathbb{P}\left(\|\widehat{\Sigma}^e - \Sigma^0\|_\infty \geq \nu\lambda_{\max}(\Sigma^0)\right) &\leq \mathbb{P}\left(\left\|\frac{1}{n-p_0}\tilde{E}\tilde{E} - \Sigma^0\right\|_\infty \geq \frac{(n\nu - p_0)\lambda_{\max}(\Sigma^0)}{n - p_0}\right) \\
&\leq 2\exp\left(-(n-p_0)\frac{\frac{(n\nu - p_0)^2}{(n-p_0)^2}}{9 + 8\frac{n\nu - p_0}{n - p_0}} + 2\log q\right) \\
&\leq 2\exp\left(-n\frac{(\nu - p_0/n)^2}{9 + 8(\nu - p_0/n)} + 2\log q\right)
\end{aligned}
$$

Now, similar to (S48), by letting $\nu = \frac{p_0}{n} + 18(c+2)\sqrt{\frac{\log q}{n}}$, we obtain that,

$$
\mathbb{P}\left(\|\widehat{\Sigma}^e - \Sigma^0\|_\infty \geq \left(\frac{p_0}{n} + 18(c+2)\sqrt{\frac{\log q}{n}}\right)\lambda_{\max}(\Sigma^0)\right) \leq 2\exp(-c\log q)
$$

when $C > 0$ and $\frac{\log q}{n} < 1$. This completes the proof of Lemma 2.

**Proof of Lemma 3.** The proof combines the proof of Theorem 1 and Lemma B.4 of Zhu and Li [2018]. Following exactly the same arguments used in Theorem 1 of Zhu and Li [2018] when $K = 1$, we know that if

$$
\begin{aligned}
\lambda &\geq 2(1 + \gamma_1^2\gamma_2)\|\widetilde{\Sigma} - \Sigma^0\|_\infty \\
a\lambda &\leq \min_{(i,j)\in A_0}|\omega_{ij}^0| - 2\gamma_2\|\widetilde{\Sigma} - \Sigma^0\|_\infty \qquad\qquad\text{(S49)} \\
\sqrt{2a} &\geq \lambda_{\max}(\Omega^0) + 2\gamma_2 q_0\|\widetilde{\Sigma} - \Sigma^0\|_\infty
\end{aligned}
$$

then (14) is a convex problem, and its minimizer $\widetilde{\Omega}$ coincides with the oracle estimator $\widehat{\Omega}^{\mathcal{O}}$. Easily, (S49) holds when $(a, \lambda)$ satisfies (S11) and (S12), and $\widetilde{\Sigma}$ satisfies (S14). Moreover, by using Lemma B.4 of Zhu and Li [2018], we have that on the event that

$$
\left\{\|\widetilde{\Sigma} - \Sigma^0\|_\infty \leq \frac{1}{2\gamma_1\gamma_2(1 + 2\gamma_1^2\gamma_2)q_0}\right\}
$$

we have

$$
\|\widehat{\Omega}^{\mathcal{O}} - \Omega^0\|_\infty \leq 2\gamma_2\|\widetilde{\Sigma} - \Sigma^0\|_\infty.
$$

Moreover, let $\widehat{\Delta} = \widehat{\Omega}^{\mathcal{O}} - \Omega^0$. Using a Taylor's expansion, we have that

$$
\begin{aligned}
\left\|[\widehat{\Omega}^{\mathcal{O}}]^{-1} - \Sigma^0\right\|_\infty &= \left\|(\widehat{\Delta} + \Omega^0)^{-1} - [\Omega^0]^{-1}\right\|_\infty \\
&= \left\|\sum_{j=1}^\infty \Sigma^0(-\widehat{\Delta}\Sigma^0)^j\right\|_\infty \le \gamma_1^2\|\widehat{\Delta}\|_\infty + \left\|R(\widehat{\Delta})\right\|_\infty \\
&\le 2\gamma_1^2\gamma_2\|\widetilde{\Sigma} - \Sigma^0\|_\infty + \|\widetilde{\Sigma} - \Sigma^0\|_\infty = (1 + 2\gamma_1^2\gamma_2)\|\widetilde{\Sigma} - \Sigma^0\|_\infty,
\end{aligned}
$$

where we have used the fact that $\left\|R(\widehat{\Delta})\right\|_\infty \le \|\widetilde{\Sigma} - \Sigma^0\|_\infty$ by an argument used in Lemma B.4 of Zhu and Li [2018]. Since $\widetilde{\Omega} = \widehat{\Omega}^{\mathcal{O}}$, thit completes the proof of (S13).

**Proof of Lemma 4.** By applying Theorem 6.1 of [Bühlmann and Van De Geer, 2011], we have that, on the event that

$$
\mathcal{J}_i = \left\{\frac{2}{n}\|X^\top E_{\cdot i}\|_\infty \le \nu_i\right\}
$$

with $\nu_i = 2\sqrt{\Sigma_{ii}^0 \frac{t^2 + 2\log p}{n}}$, the lasso estimator satisfies that

$$
\frac{1}{n}\|X(\widetilde{C}_{\cdot i} - C_{\cdot i}^0)\|_2^2 + \lambda_B\|\widetilde{C}_{\cdot i} - C_{\cdot i}^0\|_1 \le 4\lambda_B^2 p_0/\phi_0^2\,, i = 1, \ldots, q
$$

for $\lambda_B \ge 2\nu_i$, provided that a *compatibility condition* is satisfied for the design matrix:

$$
\frac{1}{n}\|X\beta\|_2^2 \ge \frac{\phi_0^2}{p_0}\|\beta\|_1^2 \text{ for all } \beta \in \mathbb{R}^p \text{ such that } \|\beta_{S_0^c}\|_1 \le 3\|\beta_{S_0}\|_1 \,.
$$

for some constant $\phi_0$. Easily, by choosing $\beta_{S_0} = 0$, we know that

$$
\phi_0^2 \le p_0\lambda_{\min}\left(\frac{1}{n}X_{S_0}^\top X_{S_0}\right) . \tag{S50}
$$

Let $\lambda_B = 2\max_i \nu_i$. Hence, on the event $\mathcal{J} := \cap_{i=1}^q \mathcal{J}_i$,

$$
\max_{1\le i\le q} \frac{1}{n}\|X(\widetilde{C}_{\cdot i} - C_{\cdot i}^0)\|_2^2 \le 16 \max_{1\le i\le q} \nu_i^2 \frac{p_0}{\phi_0^2} \text{ and } \max_{1\le i\le q} \|\widetilde{C}_{\cdot i} - C_{\cdot i}^0\|_1 \le 8 \max_{1\le i\le q} \nu_i \frac{p_0}{\phi_0^2} \tag{S51}
$$

Note that

$$\mathbb{P}(\mathcal{J}) = \mathbb{P}(\cap_{i=1}^q \mathcal{J}_i) \geq 1 - \sum_{i=1}^q \mathbb{P}(\mathcal{J}_i^c) \geq 1 - 2q\exp(-t^2/2)\,.$$

where we have used the fact that $\mathbb{P}(\mathcal{J}_i^c) \leq 2\exp(-t^2/2)$ by Lemma 6.2 of Bühlmann and Van De Geer [2011]. By letting $t = \sqrt{2(c+1)\log q}$ and $\nu_i = \sqrt{\Sigma_{ii}^0 \frac{t^2 + 2\log p}{n}} = \sqrt{2\Sigma_{ii}^0 \frac{(c+1)\log q + \log p}{n}}$, we obtain that the two inequalities in (S51) hold with probability at least $\mathbb{P}(\mathcal{J}) \geq 1 - 2\exp(-c\log q)$. This proves (S21). Consequently, on the event $\mathcal{J}$,

$$\begin{aligned}
\max_{1\leq i\leq q} \tfrac{1}{n}\|X(\widetilde{C}_{\cdot i} - C_{\cdot i}^0)\|_2^2 &\leq \tfrac{32(c+1)\max_i \Sigma_{ii}^0}{\phi_0^2} \cdot \tfrac{p_0 \log(pq)}{n}\,, \\
\max_{1\leq i\leq q} \|\widetilde{C}_{\cdot i} - C_{\cdot i}^0\|_1 &\leq \tfrac{8\sqrt{2(c+1)\max_i \Sigma_{ii}^0}}{\phi_0^2} \sqrt{\tfrac{p_0^2 \log(pq)}{n}}\,.
\end{aligned} \tag{S52}$$

Using this, we first bound $\|\widetilde{\Sigma} - \Sigma^0\|_\infty$. Note that

$$\begin{aligned}
\|\widetilde{\Sigma} - \Sigma^0\|_\infty &= \frac{1}{n}\|(X(C^0 - \widetilde{C}) + E)^\top (X(C^0 - \widetilde{C}) + E)\|_\infty \\
&\leq \left\|\frac{1}{n}E^\top E - \Sigma^0\right\|_\infty + \frac{1}{n}\|(\widetilde{C} - C^0)^\top X^\top X(\widetilde{C} - C^0)\|_\infty + \frac{2}{n}\|E^\top X(\widetilde{C} - C^0)\|_\infty
\end{aligned}$$

Next, we bound the last two terms above, respectively. On the event $\mathcal{J}$, we have that

$$\begin{aligned}
\left\|\frac{1}{n}(\widetilde{C} - C^0)^\top X^\top X(\widetilde{C} - C^0)\right\|_\infty &\leq \max_{1\leq j,j'\leq q} \frac{1}{n}\left\|X(\widetilde{C}_{\cdot j} - C_{\cdot j}^0)\right\|_2 \left\|X(\widetilde{C}_{\cdot j'} - C_{\cdot j'}^0)\right\|_2 \\
&\leq 16\max_{1\leq i\leq q}\nu_i^2 \frac{p_0}{\phi_0^2}
\end{aligned}$$

where we have used (S51) and the Cauchy-Schwarz inequality. For the last term, again applying (S51), we have

$$\begin{aligned}
\left\|\frac{2}{n}E^\top X(\widetilde{C} - C^0)\right\|_\infty &= \frac{2}{n}\max_{j,j'} e_j^\top X(\widetilde{C}_{\cdot j'} - C_{\cdot j'}^0) \\
&\leq \frac{2}{n}\max_j \|X^\top e_j\|_\infty \cdot \max_{j'}\|\widetilde{C}_{\cdot j'} - C_{\cdot j'}^0\|_1 \leq \max_i \nu_i \cdot \max_{j'}\|\widetilde{C}_{\cdot j'} - C_{\cdot j'}^0\|_1 \\
&\leq 8\max_i \nu_i^2 \frac{p_0}{\phi_0^2}
\end{aligned}$$

on the event $\mathcal{J}$. Hence, on the event $\mathcal{J}$, we have that

$$\|\widetilde{\Sigma} - \Sigma^0\|_\infty \leq \|\widehat{\Sigma}^s - \Sigma^0\|_\infty + 24 \max_i \nu_i^2 \frac{p_0}{\phi_0^2}$$

$$= \|\widehat{\Sigma}^s - \Sigma^0\|_\infty + \frac{48(c+1)\max_i \Sigma_{ii}^0}{\phi_0^2} \cdot \frac{p_0 \log(pq)}{n} \qquad (S53)$$

Then, by applying Lemma 3, we have that $\|\widetilde{\Omega} - \Omega^0\|_\infty \leq 2\gamma_2 \|\widetilde{\Sigma} - \Sigma^0\|_\infty$ on the event that

$$\left\{ \|\widetilde{\Sigma} - \Sigma^0\|_\infty \leq \min \left( \frac{\min_{(i,j)\in A_0} |\omega_{ij}^0|}{2\gamma_2 + 2a(1 + \gamma_1^2 \gamma_2)}, \frac{\sqrt{2a} - \lambda_{\max}(\Omega^0)}{2\gamma_2 q_0}, \frac{1}{2\gamma_1 \gamma_2 (1 + 2\gamma_1^2 \gamma_2) q_0} \right) \right\},$$

By choosing $a \geq \frac{(\lambda_{\max}(\Omega^0) + \gamma_1^{-1}(1 + 2\gamma_1^2 \gamma_2)^{-1})^2}{2}$ so that $\frac{\sqrt{2a} - \lambda_{\max}(\Omega^0)}{2\gamma_2 q_0} \geq \frac{1}{2\gamma_1 \gamma_2 (1 + 2\gamma_1^2 \gamma_2) q_0}$, the above event simplifies to

$$\left\{ \|\widetilde{\Sigma} - \Sigma^0\|_\infty \leq \min \left( \frac{\min_{(i,j)\in A_0} |\omega_{ij}^0|}{2\gamma_2 + 2a(1 + \gamma_1^2 \gamma_2)}, \frac{1}{2\gamma_1 \gamma_2 (1 + 2\gamma_1^2 \gamma_2) q_0} \right) \right\},$$

Combining this with (S53), we know that if

$$\frac{48(c+1)\max_i \Sigma_{ii}^0}{\phi_0^2} \cdot \frac{p_0 \log(pq)}{n} \leq \frac{1}{2} \min \left( \frac{\min_{(i,j)\in A_0} |\omega_{ij}^0|}{2\gamma_2 + 2a(1 + \gamma_1^2 \gamma_2)}, \frac{1}{2\gamma_1 \gamma_2 (1 + 2\gamma_1^2 \gamma_2) q_0} \right)$$

then we have

$$\|\widetilde{\Omega} - \Omega^0\|_\infty \leq 2\gamma_2 \|\widetilde{\Sigma} - \Sigma^0\|_\infty \qquad (S54)$$

on the event that

$$\mathcal{K} \subseteq \left\{ \|\widehat{\Sigma}^s - \Sigma^0\|_\infty \leq \frac{1}{4} \min \left( \frac{\min_{(i,j)\in A_0} |\omega_{ij}^0|}{\gamma_2 + a(1 + \gamma_1^2 \gamma_2)}, \frac{1}{\gamma_1 \gamma_2 (1 + 2\gamma_1^2 \gamma_2) q_0} \right) \right\}.$$

Combining this with the bound in (S53), we obtain that

$$\|\widetilde{\Omega} - \Omega^0\|_\infty \leq 2\gamma_2 \|\widehat{\Sigma}^s - \Sigma^0\|_\infty + \frac{96(c+1)\gamma_2 \max_i \Sigma_{ii}^0}{\phi_0^2} \cdot \frac{p_0 \log(pq)}{n}$$

on the event $\mathcal{J} \cap \mathcal{K}$. This completes the proof of (S19).

Now define $\widetilde{B} = \widetilde{C}\widetilde{\Omega}$. Then,

$$
\begin{aligned}
\|\widetilde{B}_{i\cdot} - B_{i\cdot}^0\|_2 &= \|\widetilde{C}_{i\cdot}\widetilde{\Omega} - C_{i\cdot}^0\Omega^0\|_2 \\
&\leq \|(\widetilde{C}_{i\cdot} - C_{i\cdot}^0)\Omega^0\|_2 + \|(\widetilde{C}_{i\cdot} - C_{i\cdot}^0)(\widetilde{\Omega} - \Omega^0)\|_2 + \|C_{i\cdot}^0(\widetilde{\Omega} - \Omega^0)\|_2 \\
&\leq \sqrt{\lambda_{\max}(\Omega^0)}\|\widetilde{C}_{i\cdot} - C_{i\cdot}^0\|_2 + \|(\widetilde{C}_{i\cdot} - C_{i\cdot}^0)(\widetilde{\Omega} - \Omega^0)\|_2 + \|C_{i\cdot}^0(\widetilde{\Omega} - \Omega^0)\|_2 \\
&\leq \sqrt{\lambda_{\max}(\Omega^0)}\|\widetilde{C}_{i\cdot} - C_{i\cdot}^0\|_2 + q_0\|\widetilde{\Omega} - \Omega^0\|_\infty(\|\widetilde{C}_{i\cdot} - C_{i\cdot}^0\|_2 + \|C_{i\cdot}^0\|_2) \qquad \text{(S55)}
\end{aligned}
$$

where $q_0$ is the row-wise sparsity of $\Omega^0$, and the last inequality uses the fact that for any $q_0$-sparse symmetric matrix $\Delta$, we have that

$$
\begin{aligned}
\|\Delta u\|_2 &= \sqrt{\sum_{i=1}^q (\delta_i^\top u)^2} \leq \sqrt{\sum_{i=1}^q \left(\sum_{(i,j)\in A} \delta_{ij}^2\right)\left(\sum_{j:(i,j)\in A} u_j^2\right)} \\
&\leq \sqrt{q_0}\|\Delta\|_\infty\sqrt{\sum_{i=1}^q \sum_{j:(i,j)\in A} u_j^2} \leq q_0\|\Delta\|_\infty\|u\|_2
\end{aligned}
$$

where $A = \{(i,j): \Delta_{ij} \neq 0\}$. Moreover, by using (S52), we obtain that, on the event $\mathcal{J}$

$$
\max_i \|\widetilde{C}_{i\cdot} - C_{i\cdot}^0\|_2 \leq \sqrt{q}\max_j \|\widetilde{C}_{\cdot j} - C_{\cdot j}^0\|_1 \leq \frac{8\sqrt{2(c+1)\max_i \Sigma_{ii}^0}}{\phi_0^2}\sqrt{\frac{p_0^2 q\log(pq)}{n}} \qquad \text{(S56)}
$$

Combining this with (S53) and (S55), we obtain that, on the event $\mathcal{J}$,

$$
\begin{aligned}
&\|\widetilde{B}_{i\cdot} - B_{i\cdot}^0\|_2 \\
&\leq \left(\sqrt{\lambda_{\max}(\Omega^0)} + q_0\|\widetilde{\Omega} - \Omega^0\|_\infty\right)\max_i \|\widetilde{C}_{i\cdot} - C_{i\cdot}^0\|_2 + q_0\|\widetilde{\Omega} - \Omega^0\|_\infty\|C_{i\cdot}^0\|_2 \\
&\leq \left(\sqrt{\lambda_{\max}(\Omega^0)} + 2q_0\gamma_2\|\widetilde{\Sigma} - \Sigma^0\|_\infty\right)\max_i \|\widetilde{C}_{i\cdot} - C_{i\cdot}^0\|_2 + q_0\|\widetilde{\Omega} - \Omega^0\|_\infty\|C_{i\cdot}^0\|_2 \\
&\leq 2\sqrt{\lambda_{\max}(\Omega^0)}\max_i \|\widetilde{C}_{i\cdot} - C_{i\cdot}^0\|_2 + q_0\|\widetilde{\Omega} - \Omega^0\|_\infty\|C_{i\cdot}^0\|_2 \\
&\leq \frac{16\sqrt{(c+1)\kappa(\Sigma^0)}}{\phi_0^2}\sqrt{\frac{p_0^2 q\log(pq)}{n}} + q_0\|\widetilde{\Omega} - \Omega^0\|_\infty\|C_{i\cdot}^0\|_2\,, \qquad \text{(S57)}
\end{aligned}
$$

where we have used the fact that

$$
\begin{aligned}
2q_0\gamma_2\|\widetilde{\Sigma} - \Sigma^0\|_\infty &\leq 2q_0\gamma_2\|\widehat{\Sigma}^s - \Sigma^0\|_\infty + \frac{96(c+1)\gamma_2 \max_i \Sigma_{ii}^0}{\phi_0^2} \frac{p_0q_0\log(pq)}{n} \\
&\leq \frac{1}{2}\sqrt{\lambda_{\max}(\Omega^0)} + \frac{1}{2}\sqrt{\lambda_{\max}(\Omega^0)} = \sqrt{\lambda_{\max}(\Omega^0)}
\end{aligned}
$$

holds on the event $\mathcal{K}$ and when

$$
\frac{96(c+1)\gamma_2 \max_i \Sigma_{ii}^0}{\phi_0^2} \frac{p_0q_0\log(pq)}{n} \leq \frac{1}{2}\sqrt{\lambda_{\max}(\Omega^0)}
$$

which is ensured by (S5).

Finally, we bound $\mathbb{P}(\mathcal{K})$. By using (S9) of Lemma 2, we have that if

$$
\frac{1}{4}\min\left(\frac{\min_{(i,j)\in A_0}|\omega_{ij}^0|}{\gamma_2 + a(1 + \gamma_1^2\gamma_2)}, \frac{1}{\gamma_1\gamma_2(1 + 2\gamma_1^2\gamma_2)q_0}\right) \geq 18(c+2)\sqrt{\frac{\log q}{n}} \tag{S58}
$$

then $\mathbb{P}(\mathcal{K}) \geq 1 - \exp(-c\log q)$. Note that, the above condition (S58) is ensured by (S3) in Assumption (A2), and (S6) in Assumption (A3). This proves (S22). This completes the proof of Lemma 4.

**Proof of Lemma 5:** Let $\widehat{\Sigma}_{XY} = \frac{1}{n}X^\top Y$, $\widehat{\Sigma}_{XX} = \frac{1}{n}X^\top X$, and $\widehat{\Sigma}_{YY} = \frac{1}{n}Y^\top Y$. The gradient of the negative log-likelihood can be expressed as

$$
\begin{aligned}
\nabla_B l_n(\Omega, B) &= -2\widehat{\Sigma}_{XY} + 2\widehat{\Sigma}_{XX}B\Omega^{-1} \\
\nabla_\Omega l_n(\Omega, B) &= \widehat{\Sigma}_{YY} - \Omega^{-1} - \Omega^{-1}B^\top\widehat{\Sigma}_{XX}B\Omega^{-1}
\end{aligned} \tag{S59}
$$

The oracle estimator $(\widehat{B}^{\mathcal{O}}, \widehat{\Omega}^{\mathcal{O}})$ must satisfy the following score equations

$$
\begin{aligned}
&-2X_{S_0}^\top Y + 2X_{S_0}^\top X_{S_0}\widehat{B}_{S_0}^{\mathcal{O}}[\widehat{\Omega}^{\mathcal{O}}]^{-1} = 0 \\
&\text{vec}_{A_0}\left(\widehat{\Sigma}_{YY} - [\widehat{\Omega}^{\mathcal{O}}]^{-1} - [\widehat{\Omega}^{\mathcal{O}}]^{-1}B^\top\widehat{\Sigma}_{XX}\widehat{B}[\widehat{\Omega}^{\mathcal{O}}]^{-1}\right) = 0.
\end{aligned}
$$

Hence, $\widehat{B}_{S_0} = (X_{S_0}^\top X_{S_0})^{-1} X_{S_0}^\top Y \widehat{\Omega}^{\mathcal{O}}$. In terms of the old parameterization, we have that $\widehat{B}_{S_0} = \widehat{C}_{S_0} \widehat{\Omega}^{\mathcal{O}}$ with $\widehat{C}_{S_0} = (X_{S_0}^\top X_{S_0})^{-1} X_{S_0}^\top Y$, which is the MLE of $C$ over $S_0$. This further implies that the oracle estimator $\widehat{\Omega}^{\mathcal{O}}$ must be the solution of a profile likelihood minimization problem after profiling out $C$, which is

$$\widehat{\Omega}^{\mathcal{O}} = \underset{\Omega:\, \omega_{ij}=0 \text{ for all } (i,j)\in A_0^c}{\arg\min} l_n(\Omega, \widehat{C}^{\mathcal{O}}) = \underset{\Omega:\, \omega_{ij}=0 \text{ for all } (i,j)\in A_0^c}{\arg\min} \operatorname{tr}(\widehat{\Sigma}^e \Omega) - \log\det\Omega \qquad \text{(S60)}$$

where $\widehat{\Sigma}^e = (1/n) \cdot Y^\top (I - P_{X_{S_0}}) Y = \frac{1}{n}\|(I - P_{X_{S_0}})E\|_F^2$, which is essentially the sample covariance matrix for $\Sigma^0$ based on the residual. Here we have used the fact that

$$
\begin{aligned}
l_n(\Omega, \widehat{C}^{\mathcal{O}}) &= \operatorname{tr}(\Omega (Y - X\widehat{C}^{\mathcal{O}})^\top (Y - X\widehat{C}^{\mathcal{O}})) - \log\det\Omega \\
&= \operatorname{tr}(\Omega Y^\top (I - P_{X_{S_0}})^\top (I - P_{X_{S_0}})Y) - \log\det\Omega = \operatorname{tr}(\widehat{\Sigma}^e \Omega) - \log\det\Omega.
\end{aligned}
$$

To bound $\|\nabla_B l_n(\widehat{\Omega}^{\mathcal{O}}, \widehat{B}^{\mathcal{O}})\|_\infty$ and $\|\nabla_\Omega l_n(\widehat{\Omega}^{\mathcal{O}}, \widehat{B}^{\mathcal{O}})\|_\infty$, we first simplify the gradient expression (S59). To this end, note that

$$\widehat{\Sigma}_{XY} = \frac{1}{n} X^\top (XC^0 + E) = \frac{1}{n} X^\top (XB^0 \Sigma^0 + E)$$

Using these, we have that $\nabla_{B_{S_0}} l_n(\widehat{\Omega}^{\mathcal{O}}, \widehat{B}^{\mathcal{O}}) = 0$ since $\widehat{B}_{S_0}^{\mathcal{O}}$ is the MLE over $S_0$. Moreover,

$$
\begin{aligned}
\nabla_{B_{S_0^c}} l_n(\widehat{\Omega}^{\mathcal{O}}, \widehat{B}^{\mathcal{O}}) &= -\frac{2}{n} X_{S_0^c}^\top Y + \frac{2}{n} X_{S_0^c}^\top X \widehat{C}^{\mathcal{O}} \\
&= -\frac{2}{n} X_{S_0^c}^\top (X_{S_0} C_{S_0}^0 + E) + \frac{2}{n} X_{S_0^c}^\top X_{S_0} (X_{S_0}^\top X_{S_0})^{-1} X_{S_0}^\top Y \\
&= \frac{2}{n} X_{S_0^c}^\top P_{X_{S_0}} E - \frac{2}{n} X_{S_0^c}^\top E = -\frac{2}{n} X_{S_0^c}^\top (I - P_{X_{S_0}}) E \qquad \text{(S61)}
\end{aligned}
$$

For $\nabla_\Omega l_n(\widehat{\Omega}^{\mathcal{O}}, \widehat{B}^{\mathcal{O}})$, note that

$$
\begin{aligned}
\nabla_\Omega l_n(\widehat{\Omega}^{\mathcal{O}}, \widehat{B}^{\mathcal{O}}) &= \widehat{\Sigma}_{YY} - [\widehat{\Omega}^{\mathcal{O}}]^{-1} - [\widehat{C}^{\mathcal{O}}]^\top \widehat{\Sigma}_{XX} \widehat{C}^{\mathcal{O}} \\
&= \frac{1}{n} Y^\top Y - [\widehat{\Omega}^{\mathcal{O}}]^{-1} - \frac{1}{n} \|P_{X_{S_0}} Y\|_F^2 \\
&= \frac{1}{n} \|(I - P_{X_{S_0}})(X_{S_0} C^0 + E)\|_F^2 - [\widehat{\Omega}^{\mathcal{O}}]^{-1} \\
&= \frac{1}{n} \|(I - P_{X_{S_0}}) E\|_F^2 - [\widehat{\Omega}^{\mathcal{O}}]^{-1} = \widehat{\Sigma}^e - [\widehat{\Omega}^{\mathcal{O}}]^{-1} ,
\end{aligned}
$$

where $P_{X_{S_0}}$ is a projection matrix in $\mathbb{R}^n$ onto the column space of $X_{S_0}$, and we have used the fact that

$$
X\widehat{C}^{\mathcal{O}} = X_{S_0} \widehat{C}^{\mathcal{O}}_{S_0} = X_{S_0}(X_{S_0}^\top X_{S_0})^{-1} X_{S_0}^\top Y = P_{X_{S_0}} Y .
$$

In summary, we have that

$$
\begin{aligned}
\nabla_B l_n(\widehat{\Omega}, \widehat{B}) &= -\frac{2}{n} \left( \mathbf{0}, X_{S_0^c}^\top (I - P_{X_{S_0}}) E \right) \\
\nabla_\Omega l_n(\widehat{\Omega}, \widehat{B}) &= \widehat{\Sigma}^e - [\widehat{\Omega}^{\mathcal{O}}]^{-1} .
\end{aligned}
$$

By Lemma 3,

$$
\|\widehat{\Omega}^{\mathcal{O}} - \Omega^0\|_\infty \le 2\gamma_2 \|\widehat{\Sigma}^e - \Sigma^0\|_\infty \text{ and } \left\|[\widehat{\Omega}^{\mathcal{O}}]^{-1} - \Sigma^0\right\|_\infty \le (1 + 2\gamma_1^2 \gamma_2)\|\widehat{\Sigma}^e - \Sigma^0\|_\infty \tag{S62}
$$

on the event that $\mathcal{G}$, because $\mathcal{G} \subseteq \left\{ \|\widehat{\Sigma}^e - \Sigma^0\|_\infty \le \frac{1}{\gamma_1 \gamma_2 (1 + 2\gamma_1^2 \gamma_2) q_0} \right\}$. This proves (S23). Moreover, by the second bound in (S62), we have

$$
\begin{aligned}
\left\|\nabla_\Omega l_n(\widehat{\Omega}^{\mathcal{O}}, \widehat{B}^{\mathcal{O}})\right\|_\infty &\le \left\|\widehat{\Sigma}^e - [\widehat{\Omega}^{\mathcal{O}}]^{-1}\right\|_\infty \le \|\widehat{\Sigma}^e - \Sigma^0\|_\infty + \|\Sigma^0 - [\widehat{\Omega}^{\mathcal{O}}]^{-1}\|_\infty \\
&\le 2(1 + \gamma_1^2 \gamma_2)\|\widehat{\Sigma}^e - \Sigma^0\|_\infty
\end{aligned}
$$

on event $\mathcal{G}$, which proves (S24).

Next, note that

$$
\begin{aligned}
\widehat{C}_{S_0}^{\mathcal{O}} - C_{S_0}^0 &= (X_{S_0}^\top X_{S_0})^{-1} X_{S_0}^\top Y - C_{S_0}^0 = (X_{S_0}^\top X_{S_0})^{-1} X_{S_0}^\top (XC^0 + E) - C_{S_0}^0 \\
&= (X_{S_0}^\top X_{S_0})^{-1} X_{S_0}^\top E = \frac{1}{n} \left( \frac{1}{n} X_{S_0}^\top X_{S_0} \right)^{-1} X_{S_0}^\top E
\end{aligned}
$$

By assumption,

$$
\begin{aligned}
\max_{i \in S_0} \|\widehat{C}_{i\cdot}^{\mathcal{O}} - C_{i\cdot}^0\|_2 &= \frac{1}{n} \left\| \left( \frac{1}{n} X_{S_0}^\top X_{S_0} \right)^{-1} X_{S_0}^\top E \right\|_{\infty,2} \\
&\leq \sqrt{(c+1) \frac{\max_i \Sigma_{ii}^0}{\lambda_{\min}\left( \frac{1}{n} X_{S_0}^\top X_{S_0} \right)} \frac{q \log(p_0 q)}{n}}
\end{aligned}
\tag{S63}
$$

on event $\mathcal{W}_1$. For $\|\widehat{B}_{i\cdot}^{\mathcal{O}} - B_{i\cdot}^0\|_2$, using (S63) and the same arguments used in (S57), we obtain that

$$
\begin{aligned}
\|\widehat{B}_{i\cdot}^{\mathcal{O}} - B_{i\cdot}^0\|_2 &\leq \sqrt{\lambda_{\max}(\Omega^0)} \|\widehat{C}_{i\cdot}^{\mathcal{O}} - C_{i\cdot}^0\|_2 + q_0 \|\widehat{\Omega}^{\mathcal{O}} - \Omega^0\|_\infty \left( \|\widehat{C}_{i\cdot}^{\mathcal{O}} - C_{i\cdot}^0\|_2 + \|C_{i\cdot}^0\|_2 \right) \\
&\leq (\sqrt{\lambda_{\max}(\Omega^0)} + 2\gamma_2 q_0 \|\widehat{\Sigma}^e - \Sigma^0\|_\infty) \|\widehat{C}_{i\cdot}^{\mathcal{O}} - C_{i\cdot}^0\|_2 + q_0 \|\widehat{\Omega}^{\mathcal{O}} - \Omega^0\|_\infty \|C_{i\cdot}^0\|_2 \\
&\leq 3\sqrt{\lambda_{\max}(\Omega^0)} \max_i \|\widehat{C}_{i\cdot}^{\mathcal{O}} - C_{i\cdot}^0\|_2 + q_0 \|\widehat{\Omega}^{\mathcal{O}} - \Omega^0\|_\infty \|C_{i\cdot}^0\|_2 \\
&\leq 3\sqrt{\frac{(c+1)\kappa(\Sigma^0)}{\lambda_{\min}\left( \frac{1}{n} X_{S_0}^\top X_{S_0} \right)} \frac{q \log(p_0 q)}{n}} + q_0 \|\widehat{\Omega}^{\mathcal{O}} - \Omega^0\|_\infty \|C_{i\cdot}^0\|_2
\end{aligned}
$$

on the event that $\mathcal{G} \cap \mathcal{W}_1$. This completes the proof of (S25).

Lastly, on the event that $\mathcal{W}_2$, it follows that

$$
\begin{aligned}
\max_{1 \leq i \leq q} \left\| \nabla_{B_{i\cdot}} l_n(\widehat{\Omega}^{\mathcal{O}}, \widehat{B}^{\mathcal{O}}) \right\|_2 &= \frac{2}{n} \max_{1 \leq i \leq q} \left\| X_{\cdot i}^\top (I - P_{X_{S_0}}) E \right\|_2 \leq \frac{2}{n} \left\| X^\top (I - P_{X_{S_0}}) E \right\|_{\infty,2} \\
&\leq 2\sqrt{(c+1) \max_i \Sigma_{ii}^0 \frac{q \log(pq)}{n}}
\end{aligned}
$$

from which (S26) follows.

Next, we bound $\mathbb{P}(\mathcal{W})$. Note that $\mathbb{P}(\mathcal{W}) \geq 1 - \mathbb{P}(\mathcal{W}_1^c) - \mathbb{P}(\mathcal{W}_2^c)$. We bound $\mathbb{P}(\mathcal{W}_1^c)$ and $\mathbb{P}(\mathcal{W}_2^c)$,

respectively. First note that for any vector $v \in \mathbb{R}^n$ and $t > 0$, we have that

$$
\begin{aligned}
\mathbb{P}(n^{-1}\|v^\top E\|_2 \geq \sqrt{q}t) &\leq \mathbb{P}\left(n^{-1}\sqrt{\sum_{j=1}^{q}(v^\top E_{\cdot j})^2} \geq \sqrt{q}t\right) \\
&\leq \mathbb{P}\left(n^{-1}\max_{1\leq j\leq q}|v^\top E_{\cdot j}| \geq t\right) \leq 2q\max_{1\leq j\leq q}\mathbb{P}\left(n^{-1}v^\top E_{\cdot j} \geq t\right) \\
&\leq 2q\max_{1\leq j\leq q}\exp\left\{-\frac{t^2 n^2}{\|v\|_2^2 \Sigma_{jj}^0}\right\} = 2q\exp\left\{-\frac{t^2 n^2}{\|v\|_2^2 \max_j \Sigma_{jj}^0}\right\}
\end{aligned}
$$

Let $v_i$ be the $i$-th column of $X_{S_0}\left(\frac{1}{n}X_{S_0}^\top X_{S_0}\right)^{-1}$; $i = 1, \ldots, p_0$. Let

$$
t = \sqrt{(c+1)\lambda_{\min}^{-1}\left(\frac{1}{n}X_{S_0}^\top X_{S_0}\right)\max_i \Sigma_{ii}^0 \frac{\log(p_0 q)}{n}} \ .
$$

We have that

$$
\begin{aligned}
\mathbb{P}(\mathcal{W}_1^c) &\leq \mathbb{P}\left(\frac{1}{n}\max_i \|v_i^\top E\|_2 \geq \sqrt{q}t\right) \leq p_0\max_i \mathbb{P}\left(\frac{1}{n}\|v_i^\top E\|_2 \geq \sqrt{q}t\right) \\
&\leq 2p_0 q\exp\left\{-\frac{t^2 n^2}{\max_i \|v_i\|_2^2 \max_j \Sigma_{jj}^0}\right\} \\
&\leq 2p_0 q\exp\left\{-\frac{\lambda_{\min}\left(\frac{1}{n}X_{S_0}^\top X_{S_0}\right)t^2 n}{\max_i \Sigma_{ii}^0}\right\} = 2\exp\left(-W\log(p_0 q)\right)
\end{aligned}
$$

where we have used the fact that $n^{-1}\max_{1\leq i\leq p_0}\|v_i\|_2^2 = \max \mathrm{Diag}\left\{\left(\frac{1}{n}X_{S_0}^\top X_{S_0}\right)^{-1}\right\} \leq \lambda_{\min}^{-1}\left(\frac{1}{n}X_{S_0}^\top X_{S_0}\right)$ in the last inequality.

Similarly, by letting $t = \sqrt{(c+1)\max_i \Sigma_{ii}^0 \frac{\log(pq)}{n}}$ and $v_i = (I - P_{X_{S_0}})X_{\cdot i}$; $i = 1, \ldots, p$, we have that

$$
\mathbb{P}(\mathcal{W}_2^c) \leq 2\exp\left(-c\log(pq)\right)
$$

where we have used the fact that $\max_i \|v_i\|_2^2 \leq \max_i \|X_{\cdot i}\|_2^2 = n$ Combining, we obtain that

$$
\mathbb{P}(\mathcal{W}_2^c) \leq 2\exp\left(-c\log(p_0 q)\right) + 2\exp\left(-c\log(pq)\right) \tag{S64}
$$

from which (S27) follows.

Now, we bound $\mathbb{P}(\mathcal{G})$. Using Lemma 2, we know that $\mathbb{P}(\mathcal{G}) \leq 2\exp(-c\log q)$ if

$$\min\left(\frac{1}{\gamma_1\gamma_2(1+2\gamma_1^2\gamma_2)q_0}, \frac{\sqrt{\lambda_{\max}(\Omega^0)}}{\gamma_2 q_0}\right) \geq \left(\frac{p_0}{n} + 18(c+2)\sqrt{\frac{\log q}{n}}\right)\lambda_{\max}(\Sigma^0)$$

which is ensured by (S6) and (S7) in Assumption (A3). This completes the proof.

## Detailed derivations of computational algorithms

### Proximal Newton Algorithm

We denote by $Z^{(m)} = \begin{bmatrix} B^{(m)} \\ \Omega^{(m)} \end{bmatrix} \in \mathbb{R}^{(p+q)\times q}$ the solution at $m$-th Newton iteration. $\Delta^{(m)} = \begin{bmatrix} \Delta_1^{(m)} \\ \Delta_2^{(m)} \end{bmatrix}$ the search direction at $m$-th Newton step with $\Delta_1^{(m)} \in \mathbb{R}^{p\times q}$ and $\Delta_2^{(m)} \in \mathbb{R}^{q\times q}$. Again we obtain the the Newton step by solving an approximation of the original problem by replacing the smooth part of the objective function with a quadratic approximation at the current solution $Z^{(m)}$. In particular, the Newton step at $m$-th Newton iteration can be obtained by solving the following optimization problem

$$\underset{\Delta\in\mathbb{R}^{(p+q)\times q}}{\text{minimize}}\,(1/2)\cdot\mathrm{Tr}(\Delta\Sigma^{(m)}\Delta^\top H^{(m)}) - \mathrm{Tr}(\Delta^\top C^{(m)}) + p(Z^{(m)}+\Delta)\,, \tag{S65}$$

where

$$H^{(m)} = \begin{pmatrix} 2\widehat{\Sigma}_{XX} & -2\widehat{\Sigma}_{XX}B^{(m)}\Sigma^{(m)} \\ -2\Sigma^{(m)}[B^{(m)}]^\top\widehat{\Sigma}_{XX} & 2\Sigma^{(m)}[B^{(m)}]^\top\widehat{\Sigma}_{XX}B^{(m)}\Sigma^{(m)}+\Sigma^{(m)} \end{pmatrix}$$

$$C^{(m)} = \begin{pmatrix} 2\widehat{\Sigma}_{XY}-2\widehat{\Sigma}_{XX}B^{(m)}\Sigma^{(m)} \\ \Sigma^{(m)}[B^{(m)}]^\top\widehat{\Sigma}_{XX}B^{(m)}\Sigma^{(m)}+\Sigma^{(m)}-\widehat{\Sigma}_{YY} \end{pmatrix}\,,\text{ and }\Sigma^{(m)}=\left[\Omega^{(m)}\right]^{-1}$$

with $\widehat{\Sigma}_{XX} = \frac{X^\top X}{n}$, $\widehat{\Sigma}_{XY} = \frac{X^\top Y}{n}$, and $\widehat{\Sigma}_{YY} = \frac{Y^\top Y}{n}$.

Next, we do another change of variable: $Z = Z^{(m)} + \Delta$. Then, the above problem is equivalent

to the following problem

$$\underset{Z\in\mathbb{R}^{(p+q)\times q}}{\text{minimize}}\,(1/2)\cdot\text{Tr}(Z\Sigma^{(m)}Z^\top H^{(m)})-\text{Tr}(Z^\top G^{(m)})+p(Z)\,,$$

where

$$G^{(m)}=\begin{pmatrix}2\widehat{\Sigma}_{XY}-2\widehat{\Sigma}_{XX}B^{(m)}\Sigma^{(m)}\\[2mm]\Sigma^{(m)}[B^{(m)}]^\top\widehat{\Sigma}_{XX}B^{(m)}\Sigma^{(m)}+2\Sigma^{(m)}-\widehat{\Sigma}_{YY}\end{pmatrix}.$$

We propose to solve the subproblem using alternating direction methods of multipliers (ADMM). Specifically, the ADMM updates are

$$\begin{aligned}Z^{(k+1)}\quad\in\quad&\underset{Z\in\mathbb{R}^{(p+q)\times q}}{\arg\min}\left\{\frac{1}{2}\text{Tr}(Z\Sigma^{(m)}Z^\top H^{(m)})-\text{Tr}(Z^\top G^{(m)})\right.\\[2mm]&\left.+\text{Tr}\left(Z^\top(2\Gamma^{(k)}-\Gamma^{(k-1)})\right)+\frac{\rho}{2}\|Z-Z^{(k)}\|_F^2\right\}\\[2mm]\Gamma^{(k+1)}\quad=\quad&\text{prox}_{\rho p^*(\cdot)}(\Gamma^{(k)}+\rho Z^{(k+1)})\,,\end{aligned}$$

where the second update amounts to performing the projection of the first $p$ rows of $Z^{(k+1)}$ onto the $L_2$ ball and the last $q\times q$ entries onto some interval, both of which are easy to carry out. The first update, however, could be challenging to carry out, because it is equivalent a linear system involving $(p+q)\times q$ variables. Naively solving such a linear system would require $O((p+q)^3\times q^3)$ computation. Here we show that this linear system can be solved much faster due to its special structure. Toward this end, note that the linear system involving $Z^{(k+1)}$ is

$$H^{(m)}Z^{(k+1)}\Sigma^{(m)}+\rho Z^{(k+1)}=G^{(m)}+\Gamma^{(k-1)}-2\Gamma^{(k)}+\rho Z^{(k)}$$

We write the eigenvalue decomposition of $H^{(m)}$ and $\Sigma^{(m)}$ as $H^{(m)}=U_1\Lambda_1 U_1^\top$ and $\Sigma^{(m)}=U_2\Lambda_2 U_2^\top$, and plug them in the above equation, we obtain

$$\Lambda_1 U_1^\top Z^{(k+1)}U_2\Lambda_2+\rho U_1^\top Z^{(k+1)}U_2=U_1^\top\left(G^{(m)}+\Gamma^{(k-1)}-2\Gamma^{(k)}+\rho Z^{(k)}\right)U_2$$

Denote by $Z'^{(k+1)} = U_1^\top Z^{(k+1)} U_2$, we have the following explicit formula for $Z'^{(k+1)}$

$$Z_{ij}'^{(k+1)} = \frac{\left[U_1^\top \left(G^{(m)} + \Gamma^{(k-1)} - 2\Gamma^{(k)} + \rho Z^{(k)}\right) U_2\right]_{ij}}{\rho + \Lambda_1(i,i)\Lambda_2(j,j)}.$$

Then, we can update $Z^{(k+1)}$ easily through $Z^{(k+1)} = U_1 Z'^{(k+1)} U_2^\top$. Note that the first ADMM update has a $O\left(p^3 \vee q^3\right)$ computational complexity.

We terminate the algorithm based on duality gap:

$$
\begin{aligned}
&\frac{1}{2}\operatorname{Tr}((Z^{(k+1)})^\top H^{(m)} Z^{(k+1)} \Sigma^{(m)}) - \operatorname{Tr}((Z^{(k+1)})^\top G^{(m)}) + p(Z^{(k+1)}) \\
&\quad + \frac{1}{2}\operatorname{Tr}\left((G^{(m)} - \Gamma^{(k+1)})^\top [H^{(m)}]^{-1}(G^{(m)} - \Gamma^{(k+1)})[\Sigma^{(m)}]^{-1}\right) \\
&= \frac{1}{2}\operatorname{Tr}([Z'^{(k+1)}]^\top \Lambda_1 Z'^{(k+1)} \Lambda_2) - \operatorname{Tr}((Z^{(k+1)})^\top G^{(m)}) + p(Z^{(k+1)}) \\
&\quad + \frac{1}{2}\operatorname{Tr}([\Gamma'^{(k+1)}]^\top \Lambda_1^{-1} [\Gamma'^{(k+1)}]\Lambda_2^{-1}),
\end{aligned}
$$

where $\Gamma'^{(k+1)} = U_1^\top (G^{(m)} - \Gamma^{(k+1)})U_2$. At termination, we return $\widetilde{Z}^{(k+1)} = Z^{(k+1)} - \rho^{-1}(\Gamma^{(k+1)} - \Gamma^{(k)})$ to get exactly sparse solution.

**Line search for proximal Newton method**

Find the largest $0 < t \leq 1$ that satisifies the following

$$
\begin{aligned}
&l(Z^{(m)} + t\Delta^{(m)}) + p(Z^{(m)} + t\Delta^{(m)}) - l(Z^{(m)}) - p(Z^{(m)}) \\
&\leq \alpha \left(p(Z^{(m)} + t\Delta^{(m)}) - p(Z^{(m)}) - t\langle C^{(m)}, \Delta^{(m)}\rangle\right),
\end{aligned}
\tag{S66}
$$

where $0 < \alpha < 1$ is some absolute constant.

Next we show that the line search will be terminated after a finite number of steps, that is, the line search condition (S66) will be satisfied for small enough $t$. To this end, first note that for any

$0 < \alpha < 1$, we have that

$$
\begin{aligned}
& (1/2) \cdot \mathrm{Tr}(\Delta^{(m)} \Sigma^{(m)} (\Delta^{(m)})^\top H^{(m)}) + \langle \nabla l(Z^{(m)}), \Delta^{(m)} \rangle + p(Z^{(m)} + \Delta^{(m)}) \\
\leq\ & (1/2) \cdot \mathrm{Tr}((\alpha\Delta^{(m)}) \Sigma^{(m)} (\alpha\Delta^{(m)})^\top H^{(m)}) + \langle \nabla l(Z^{(m)}), \alpha\Delta^{(m)} \rangle + p(Z^{(m)} + \alpha\Delta^{(m)}) \\
\leq\ & (\alpha^2/2) \cdot \mathrm{Tr}(\Delta^{(m)} \Sigma^{(m)} (\Delta^{(m)})^\top H^{(m)}) + \alpha\langle \nabla l(Z^{(m)}), \Delta^{(m)} \rangle + \alpha p(Z^{(m)} + \Delta^{(m)}) + (1 - \alpha)p(Z^{(m)}),
\end{aligned}
$$

because $\Delta^{(m)}$ minimizes (S65). Hence,

$$
\begin{aligned}
\delta \ \equiv\ & p(Z^{(m)} + \Delta^{(m)}) - p(Z^{(m)}) + \langle \nabla l(Z^{(m)}), \Delta^{(m)} \rangle \\
\leq\ & -\frac{1 - \alpha^2}{2(1 - \alpha)} \cdot \mathrm{Tr}(\Delta^{(m)} \Sigma^{(m)} (\Delta^{(m)})^\top H^{(m)}) \\
\rightarrow\ & -\mathrm{Tr}(\Delta^{(m)} \Sigma^{(m)} (\Delta^{(m)})^\top H^{(m)})
\end{aligned}
$$

as $\alpha \to 1$. Moreover, we have that

$$
\begin{aligned}
& l(Z^{(m)} + t\Delta^{(m)}) + p(Z^{(m)} + t\Delta^{(m)}) - l(Z^{(m)}) - p(Z^{(m)}) \\
\leq\ & t\langle \Delta^{(m)}, \nabla l(Z^{(m)}) \rangle + O(t^2) + p(Z^{(m)} + t\Delta^{(m)}) - p(Z^{(m)}) \\
\leq\ & \alpha\left( p(Z^{(m)} + t\Delta^{(m)}) - p(Z^{(m)}) - t\langle \Delta^{(m)}, C^{(m)} \rangle \right)
\end{aligned}
$$

for some $0 < \alpha < 1$, where the last inequality uses the fact that

$$
\begin{aligned}
& p(Z^{(m)} + t\Delta^{(m)}) - p(Z^{(m)}) + t\langle \Delta^{(m)}, \nabla l(Z^{(m)}) \rangle \\
\leq\ & t(p(Z^{(m)} + \Delta^{(m)}) - p(Z^{(m)})) + t\langle \Delta^{(m)}, \nabla l(Z^{(m)}) \rangle \\
\leq\ & -t\,\mathrm{Tr}(\Delta^{(m)} \Sigma^{(m)} (\Delta^{(m)})^\top H^{(m)}) \leq 0
\end{aligned}
$$

Moreover, we can show that the line search condition (S66) will be satisfied for $t = 1$ when $Z^{(m)}$ is close to the solution, provided that $\nabla^2 l(\cdot)$ is Lipschitz continuous with constant $L > 0$ and $\widehat{\Sigma}_{XX}$ is strictly positive definite.

In fact, by Lipschitz continuity of $l(\cdot)$, it is easy to show that

$$l(Z^{(m)} + \Delta^{(m)})$$
$$\leq\ l(Z^{(m)}) + \langle \nabla l(Z^{(m)}), \Delta^{(m)} \rangle + (1/2) \cdot \mathrm{Tr}(\Delta^{(m)} \Sigma^{(m)} (\Delta^{(m)})^\top H^{(m)}) + \frac{L\|\Delta^{(m)}\|_F^3}{6}$$

Therefore,

$$l(Z^{(m)} + \Delta^{(m)}) + p(Z^{(m)} + \Delta^{(m)}) - l(Z^{(m)}) - p(Z^{(m)})$$
$$\leq\ \delta + (1/2) \cdot \mathrm{Tr}(\Delta^{(m)} \Sigma^{(m)} (\Delta^{(m)})^\top H^{(m)}) + \frac{L\|\Delta^{(m)}\|_F^3}{6}$$
$$\leq\ \frac{\delta}{2} + \frac{L\|\Delta^{(m)}\|_F^3}{6} \leq \alpha\delta$$

for some $0 < \alpha < 1/2$.

## Proximal gradient method details

To apply the fast proximal gradient algorithm of Beck and Teboulle [2009], we only need expressions for gradient of the smooth part of the objective function with respect to $\Omega$ and $B$, and the proximal operation for the nonsmooth part. For the gradients, it is easy to see that for the pseudo-likelihood loss,

$$\nabla_B l^{\mathrm{pseudo}}(\Omega, B)\ =\ -\tfrac{1}{n} X^\top (Y\Omega - XB)\Omega_D^{-1},$$
$$\nabla_\Omega l^{\mathrm{pseudo}}(\Omega, B)\ =\ \tfrac{1}{2n} Y^\top (Y\Omega - XB)\Omega_D^{-1} + \tfrac{1}{2n}\Omega_D^{-1}(Y\Omega - XB)^\top Y,$$
$$-\ \tfrac{1}{2n}\Omega_D^{-2}\mathrm{Diag}\Big[(Y\Omega - XB)^\top(Y\Omega - XB)\Big] - \tfrac{1}{2}\Omega_D^{-1}$$

and for the D-trace loss, we have that

$$\nabla_B l^{\mathrm{D\text{-}trace}}(\Omega, B)\ =\ -n^{-1} X^\top (Y\Omega - XB),$$
$$\nabla_\Omega l^{\mathrm{D\text{-}trace}}(\Omega, B)\ =\ \tfrac{1}{2n}\left(Y^\top (Y\Omega - XB) + (Y\Omega - XB)^\top Y\right) - I.$$

# Figures and tables

Figure S1: Three types of graphs used in our simulations.

Table S1: Estimation error of $C$ and $(C, \Omega)$ (denoted as Error$(\widehat{C})$ and Error$(\widehat{C}, \widehat{\Omega})$), false positive rate of $C$, and false negative rate of $C$, as well as their standard errors (in parentheses) of various methods over 100 simulations, where NA means not applicable. Here Wang, MRCE, our-Dtrace, our-Pseudo, and Our-ML denote the method of Wang [2015], the method of Rothman et al. [2010], the proposed method with Dtrace loss, the Pseudo likelihood loss, and the negative log-likelihood loss, respectively.

| $(p,q)$ | Method | Error$(\widehat{C})$ | Error$(\widehat{\Omega})$ | Error$(\widehat{C}, \widehat{\Omega})$ | FPR$(\widehat{C})$ | FNR$(\widehat{C})$ |
|---|---|---|---|---|---|---|
| $(100, 3)$ | Our-ML | .049(.025) | .031(.019) | .082(.034) | 0(.002) | 0(0) |
| | Our-Dtrace | .05(.024) | .033(.021) | .086(.036) | .014(.021) | 0(0) |
| | Our-Pseudo | .05(.024) | .028(.017) | .079(.031) | .012(.022) | 0(0) |
| | MRCE | .568(2.91) | .159(.254) | .523(.354) | .396(.085) | 0(0) |
| | Wang | .332(.12) | NA | NA | .041(.023) | 0(0) |
| $(200, 3)$ | Our-ML | .05(.027) | .031(.019) | .084(.037) | 0(.002) | 0(0) |
| | Our-Dtrace | .049(.023) | .033(.02) | .085(.034) | .007(.011) | 0(0) |
| | Our-Pseudo | .05(.024) | .028(.017) | .08(.032) | .007(.013) | 0(0) |
| | MRCE | 9.52(12) | 1773(5496) | 30004(110439) | .786(.234) | 0(0) |
| | Wang | .417(.125) | NA | NA | .025(.017) | 0(0) |

Table S2: Estimation error of $C$, $\Omega$, and $(C, \Omega)$, FP of $C$, and FN of $C$, as well as their standard errors (in parentheses) of various methods over 100 simulations of example with $(p, q) = (3, 100)$, where NA means not applicable. Here Wang, MRCE, our-Dtrace, our-Pseudo, and Our-ML denote the method of Wang [2015], the method of Rothman et al. [2010], the proposed method with Dtrace loss, the Pseudo likelihood loss, and the negative log-likelihood loss, respectively.

| Graph | Method | Error($\widehat{C}$) | Error($\widehat{\Omega}$) | Error($\widehat{C}, \widehat{\Omega}$) | FPR($\widehat{\Omega}$) | FNR($\widehat{\Omega}$) |
|---|---|---|---|---|---|---|
| band | Our-ML | 1.54(.144) | 1.53(.227) | 3.15(.303) | .006(.002) | .002(.004) |
| | Our-Dtrace | 1.55(.142) | 2.16(.31) | 3.74(.428) | .03(.009) | .05(.007) |
| | Our-Pseudo | 1.56(.158) | NA | NA | .007(.004) | .074(.006) |
| | MRCE | 1.55(.143) | 27.3(14.5) | 29.1(15) | .177(.342) | .74(.436) |
| | Wang | 1.61(.156) | NA | NA | .117(.006) | 0(0) |
| hub | Our-ML | 1.54(.143) | 2.51(.398) | 4.17(.433) | .018(.006) | .022(.018) |
| | Our-Dtrace | 1.54(.142) | 3.26(.385) | 4.81(.438) | .011(.01) | .034(.018) |
| | Our-Pseudo | 1.55(.144) | 3.12(.381) | 4.79(.447) | .018(.016) | .024(.023) |
| | MRCE | 1.55(.143) | 16.1(11.3) | 17.8(11.7) | .117(.276) | .598(.403) |
| | Wang | 1.61(.168) | NA | NA | .146(.005) | .001(.003) |
| random | Our-ML | 1.54(.142) | 2.53(.352) | 4.18(.386) | .012(.004) | .043(.027) |
| | Our-Dtrace | 1.54(.143) | 2.62(.372) | 4.26(.383) | .011(.005) | .050(.024) |
| | Our-Pseudo | 1.53(.142) | 2.54(.385) | 4.16(.398) | .013(.006) | .041(.026) |
| | MRCE | 1.55(.143) | 16.4(11.4) | 18.1(11.7) | .124(.293) | .8(.394) |
| | Wang | 1.6(.151) | NA | NA | .132(.006) | .001(.003) |

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

Table S3: Estimation error of $C$, $\Omega$, and $(C, \Omega)$, FP of $C$, and FN of $C$, as well as their standard errors (in parentheses) of various methods over 100 simulations of example with $(p, q) = (50, 50)$, where NA means not applicable. Here Wang, MRCE, our-Dtrace, our-Pseudo, and Our-ML denote the method of Wang [2015], the method of Rothman et al. [2010], the proposed method with Dtrace loss, the Pseudo likelihood loss, and the negative log-likelihood loss, respectively.

| Graph | Method | Error($\widehat{C}$) | Error($\widehat{\Omega}$) | Error($\widehat{C}, \widehat{\Omega}$) | FPR($\widehat{C}$) | FNR($\widehat{C}$) | FPR($\widehat{\Omega}$) | FNR($\widehat{\Omega}$) |
|---|---|---|---|---|---|---|---|---|
| band | Our-ML | .775(.108) | .812(.202) | 1.63(.245) | 0(0) | 0(0) | .009(.01) | .005(.01) |
| | Our-Dtrace | .779(.108) | .850(.254) | 1.70(.278) | .001(.007) | 0(0) | .009(.009) | .006(.006) |
| | Our-Pseudo | .792(.11) | .801(.224) | 1.62(.226) | .004(.014) | 0(0) | .009(.009) | .005(.003) |
| | MRCE | 4.52(.597) | 11.5(.112) | 15(.423) | .986(.018) | 0(0) | 0(0) | 1(0) |
| | Wang | 3.3(.433) | NA | NA | .729(.082) | 0(0) | .158(.013) | 0(.002) |
| hub | Our-ML | .776(.108) | 1.25(.225) | 2.09(.288) | 0(0) | 0(0) | .048(.028) | .016(.027) |
| | Our-Dtrace | .78(.109) | 1.32(.213) | 2.15(.274) | .001(.004) | 0(0) | .045(.021) | .02(.024) |
| | Our-Pseudo | .787(.111) | 1.21(.283) | 2.03(.219) | .001(.004) | 0(0) | .041(.026) | .021(.028) |
| | MRCE | 4.22(.618) | 7.59(.109) | 11.1(.478) | .986(.018) | 0(0) | 0(0) | 1(0) |
| | Wang | 3.12(.416) | NA | NA | .76(.074) | 0(0) | .199(.013) | .008(.013) |
| random | Our-ML | .777(.109) | 1.51(.24) | 2.36(.285) | 0(0) | 0(0) | .041(.008) | .077(.043) |
| | Our-Dtrace | .777(.109) | 1.574(.24) | 2.42(.275) | .001(.005) | 0(0) | .035(.007) | .080(.041) |
| | Our-Pseudo | .781(.109) | 1.58(.24) | 2.45(.255) | .002(.007) | 0(0) | .044(.007) | .071(.039) |
| | MRCE | 3.77(.4) | 5.8(.094) | 9.23(.388) | .989(.015) | 0(0) | 0(0) | 1(0) |
| | Wang | 3.29(.395) | NA | NA | .769(.08) | 0(0) | .18(.014) | .026(.021) |

Y. Zhu and L. Li. Multiple matrix gaussian graphs estimation. *Journal of the Royal Statistical Society, Series B.*, In press., 2018.