[Reviews · NeurIPS 2020]

Review 1

Summary and Contributions: This paper considers the problem of simultaneous estimation of the regression coefficients and the precision/error covariance matrix in high dimensional multivariate linear models, for which the traditional (maximum likelihood based) approaches typically lead to a challenging non-convex optimization problem. To resolve the challenge of non-convexity, the authors propose a new parameterization, and show that the negative log-likelihood function as well as two alternative loss functions (pseudo-likelihood and Dtrace, which are computationally more efficient) are all convex under this parameterization. In addition, the new parameterization maintains the same rank and row-sparsity as the original parameterization, which enables the application of existing regularization approaches. The authors then apply a non-convex (group) sparsity-inducing penalty and propose a Difference-of-Convex (DC) approach to solve the regularized optimization problem with the new parameterization, and show that under reasonable conditions, this approach achieves exact recovery of the oracle estimators in just one iteration with appropriate initialization. The rate of convergence in terms of the distance to the underlying true parameters is also obtained. Finally, numerical experiments are conducted and show that the proposed convex formulation and DC approach achieve improvement over existing methods by several orders of magnitude. In short, the major contribution of this paper is that it 1) proposes the first convex parameterization/formulation for the problem of simultaneous estimation of the regression coefficients and the precision matrix in multivariate linear models; 2) propose a DC approach to solve the regularized problem with a non-convex sparsity-inducing penalty with high probability exact recovery results (in terms of the oracle estimators) and convergence rate bounds (in terms of the true parameters); 3) numerical experiments show the superiority of the proposed approach.

Strengths: The major strengths of the paper are listed below: 1. This paper proposes the first convex parameterization/formulation for the problem of simultaneous estimation of the regression coefficients and the precision matrix in high-dimensional multivariate linear models. The convexity is shown for both the classical negative log-likelihood function and also two more recently proposed alternative loss functions with better computational efficiency in practice. In addition, the proposed new parameterization does not change the rank and row-sparsity of the original parameters, making it reasonable to apply existing regularization terms to the new parameterization directly. 2. The authors also propose a DC approach to solve the regularized problem with a non-convex sparsity-inducing penalty with high probability exact recovery results (in terms of the oracle estimators) and convergence rate bounds (in terms of the true parameters). 3. The authors also conduct several numerical experiments to show that the proposed convex formulation and DC approach together achieve improvement over existing algorithms by several orders of magnitude.

Weaknesses: The major weaknesses of the paper are listed below: 1. There are some potential inaccuracies in the description of the algorithm. For example, in Section 3.1, the first equalities in the two lines of equations after line 210 should be \approx instead, right? And does the notation p_{\tau_B}^’ denote the sub-gradient of p_{\tau_B}? In general, some more explanations about the linearization here would be helpful. 2. The initialization proposed in Section 4 leads to pretty strong theoretical guarantees (exact recovery of the oracle parameters in just one iteration). However, it seems that such kind of initialization is not used in the numerical experiments in Section 5. If the implementation actually includes this initialization, then the authors should state this more explicitly. Or is there any practical limitation of this initialization? If not, the authors should provide numerical results with this special initialization procedure. Otherwise, the authors may want to comment on whether such kind of initialization is also tractable and efficient in practice. 3. The theoretical results in Section 4 have some unclear parts. Firstly, in lines 244-246, the authors claim that (14) is convex for certain range of \lambda_{\Omega} and a. Does the requirement of these tuning parameters in Theorem 3 ensure that (14) is convex? Secondly, in Theorem 3, what does O_p exactly indicate? Based on the proof in the appendix, it seems that (15) and (16) also hold with probability at least 1-12\exp(-C\log q) (and with O() instead of O_p). So there seems to be some slight inconsistency. Please correct me if I missed anything here. And for the proof in the appendix, the authors should clearly explain why (S29) and (S30) ensure that the first iterate of the DC algorithm coincides with the oracle estimator when initialized by (\tilde{B},\tilde{\Omega}). On a related point, the authors should explain why (S38) is sufficient to ensure that that the DC algorithm stops at the second iterate and recovers the oracle estimator. Also, what if (S38) contradicts with (S29) and (S30)? Can we ensure that these three conditions are satisfied altogether?

Correctness: The claims and method are correct, as far as I understand. The only concerns have been mentioned in the weaknesses section above. The empirical methodology is also correct.

Clarity: Yes, the paper is generally well written and easy to follow. The concerns and suggestions about writing improvements can be found in other sections of the review.

Relation to Prior Work: Yes, the novelty of the convex parameterization and so on have been clearly placed. Some minor concerns about comparisons with existing works have also been listed in the weaknesses section above.

Reproducibility: Yes

Additional Feedback: ================== post rebuttal ==================================== I have read the authors' rebuttal and I think they have addressed my concern of DC algorithm description. In addition, the comparison with using convex penalties is also interesting and persuasive. However, the authors seem to skip two of my other main questions, one on initialization and the other on some potential mistakes and clarity issues in the theoretical results in Section 4 (and the related proofs in the appendix). Given that they are directly related to correctness and the theory-practice gap of the paper, I decide to regretfully lower my score to 6. ====================================================== The new convex parameterization is both significant and elegant, and the exact recovery and convergence rate guarantees of the DC algorithm is also clean and inspiring. In addition, the writing is good and I enjoy reading this paper in general. The only concerns about this paper lie in some writing and technical details, as mentioned above in the weaknesses section. The authors may want to address them as much as possible in the rebuttal phase. Finally, please find some additional suggestions on typo fixing, writing improvement and so on. 1. In line 12, there is an extra “of”. 2. In line 16, “outperforms” should be “outperform”. 3. In line 22, “logitudinal” should be “longitudinal”. 4. In lines 51-52, the authors may want to make it clearer whether Molstad and Rothman [2016] formulates a convex problem or not, and how it compares with the current paper (in terms of novelty). 5. In line 86, “particluar” should be “particular”. 6. In line 104, “depdence” should be “dependence”. 7. In lines 106-107, the authors should better explicitly state the full-rank assumptions on X and Y-X\hat{C}^{MLE}, which are needed for the formulae below to be well-defined. 8. In line 114, there should be a space before “For example”. 9. In line 128, “Theorem” might better be “theorem”. 10. In lines 133-135, the authors mention another parameterization that leads to a convex optimization problem. Is the convexity result of this alternative parameterization existing in the literature or something that is also discovered by the authors? I would suggest the authors to specify this more clearly to claim the deserved credit. This would also make the contribution of the new convex parameterization clearer. 11. In line 136, the “in” before “when” should be removed. 12. In line 140, “effet” should be “effect”. 13. In line 154, “is to” is weird (grammatically) and should better be rephrased. 14. In Section 2.3, the authors introduce \Sigma^0 and \Omega^0. The authors should actually introduce them as well as C^0 (which the authors seem to forget to define in the paper) at an earlier place (before Section 2.1) to make the notational reference easier. In particular, currently C_i^0 seems to be undefined in line 230. 15. In Theorem 3, the constant C in the probability 1-12\exp(-C\log q) is undefined and also conflicts with the variable C notation. 16. In line 254, \Omega should be \Omega^0. 17. In line 280, (\lambda_1,\lambda_2) should be (\lambda_B,\lambda_{\Omega}), right? 18. In the references, there are also a few typos. In line 357, “and others” should be replaced with the full author list. In line 389, “d-trace” should be “D-trace”. Similarly, in line 381, “gaussian” should be “Gaussian”.


Review 2

Summary and Contributions: A regularized multivariate multiple lineae regression framework is proposed for joint estimation of regression coefficients and error covariance matrix from high-dimensional data. Since the negative log-likelihood function is not jointly convex in regression coefficients and error covariance matrix, a new parameterization is proposed under which the negative log-likelihood function is proved to be convex. For faster computation, two other alternative loss functions are also considered, and proved to be convex under the proposed parameterization. Efficient algorithms are developed to solve the proposed penalized multivariate regression problem with the negative log-likelihood loss and the two alternative loss functions, based on difference-of-convex (DC) formulations and fast proximal methods. It derives conditions under which the proposed estimator recovers the oracle estimator, and the convergence rates. Numerical experiments with various sample sizes and dimensions show that the joint estimation approach significantly outperforms (up to 402%-1159% reduction in error) two existing methods in terms of parameter estimation and support recovery of regression coefficients and error precision matrix.

Strengths: This is a solid piece of work. The proposed models, algorithms and theories are sound and look novel. See "Summary and contributions".

Weaknesses: The clarity of presentation could be improved. Real data experiments are missing. Broader impact is not discussed.

Correctness: Yes. The theories and algorithms are sound and verified in numerical experiments.

Clarity: Yes.

Relation to Prior Work: Yes.

Reproducibility: Yes

Additional Feedback: Comments: 1. In Section 3.1, the DC algorithm is not clearly described. - In the two equations between Lines 210 and 211, the leftmost $=$ should be $\leq$, and the rightmost $=$ hold only up to additive constants. - Moreover, the DC algorithm first writes the penalty function $p_\tau(\cdot)$ in the form of the difference of two convex functions $p_\tau(\cdot) = g(\cdot) - h(\cdot)$, and then linearize $h(\cdot)$ at iteration $t$ to construct an upper approximation of $p_\tau(\cdot)$. These should be made clear. - What are the advantages in theory (in terms of convergence rate, global optimality conditions, etc.) of using a DC algorithm to optimize a DC function (e.g., $p_\tau(x)$ considered in this work), compared with a generic iterative algorithm that optimizes a non-DC function (e.g., $p(x)=|x|^p$ with $0<p\leq 1$) via a sequence of convex relaxations ($|x|^p \leq \frac{p}{2} \frac{x^2}{|x^{(t)}|^{2-p}} + \frac{2-p}{2} |x^{(t)}|^p$)? A brief discussion is useful in helping the reader better understand the DC algorithm. 2. The sample size is not specified in Section 5 ``Numerical Studies". 3. Typos: Line 12: There is a redundant ``of" in ``of of". Line 108: ``regularization(s) are". Line 114: Missing space after the full stop. Line 144: ``is only dependent to (on)". Line 151: The notation of the $i$-th row of $B$ is different from that in Section 3.1. Please be consistent. Line 167: ``For (a) truly large-scale problem". Line 179: $\mathop{{\rm Diag}}(\Omega)$ is not defined. Line 217: ``to generated" should be ``generated" or ``to be generated". The $C^0$ in Line 230 and Theorem 3 is not defined. Line 243: $p_{a,\lambda}$ should be $p_{\lambda,a}$. Lines 268-269, ``the nonzero entries in $\Omega$ encodes a (encode an) undirected graph". Line 314: There is a redundant ``the" in ``the the". Supplementary Material: Page 2, line 4: The $U^{\rm T}$ (which appears four times) should be $U$. Page 2, line 6: ``$b_i, c_i$ are the $i$-th row of $B \Omega^{-1/2} U^{\rm T}$ and $V_2 \Omega^{-1/2} U^{\rm T}$, respectively" should be ``$b_i, c_i$ are the $i$-th columns of $B \Omega^{-1/2} U$ and $V_2 \Omega^{-1/2} U$, respectively". Page 2, line 8: $\Omega + t V_1 \succeq 0$ should be $\Omega + t V_1 \succ 0$. First line below Eq. (S1): If $\phi_0=0$ than inequality (S1) always holds. To exclude this trivial case, should ``for some constant $\phi_0$" be ``for some constant $\phi_0^2 > 0$"? Section ``Proof of Theorem 3", line 3: $1 \leq i,j \leq p$ and $\sum_{j=1}^p$ should be $1 \leq i,j \leq q$ and $\sum_{j=1}^q$, respectively. Tables S1-S3: Please add the word ``respectively" at the end of the table titles. ========================= After rebuttal ========================= I have read all reviews and the authors' responses. The authors have addressed my concerns. The authors also tried to address Comments 1-4 of Reviewer #3. For example, they performed additional simulations showing the new reparameterization (a convex function) combined with a convex penalty (“Our-ML-convex” in Table 1) generally outperforms the old reparameterization (a non-convex function) using the same convex penalty (“MRCE-group” in Table 1). It may be speculated that when combined with a non-convex penalty, the new reparameterization may also outperform the old reparameterization.


Review 3

Summary and Contributions: This paper considers a multivariate regression problem and proposes a convex optimization formulation for high dimensional multivariate linear regression under error covariance structure.

Strengths: The introduced parametrization of the multivariate regression problem can be stated as a convex program, which has several computational advantages: (1) Regularization can be directly applied on the new parametrization, (2) convex optimization solvers can be employed. In addition to this, the authors provide two alternative versions with different loss functions to further accelerate the computation.

Weaknesses: 1. The novelty of Theorem 1 is not entirely clear. The authors argue that the classical another parametrization B, Theta = (C Omega^{1/2}, Omega^{1/2}), which also enables joint convexity. Therefore, Theorem 1 doesn't appear to be a significant result in itself. This may be a lemma instead. 2. It is not clear why the classical parametrization does not allow penalty functions. One can include penalties on C by using the classical parametrization and levering C=B Theta^{-1}, e.g., || B Theta^{-1}||_{1,2} where 1,2 is the group L1 penalty. 3. It looks like the authors argue the advantage of convexity in section 2.1, but later on introduce non-convex regularizers, which in my opinion destroys the purpose. The resulting overall problem is non-convex. It is very unclear what convexity of a part of the objective function provides. Moreover, without any empirical comparison with the standard parametrization, it is hard to claim an advantage. 4. This work needs a more detailed comparative analysis in order to prove the superiority of the proposed approach. In particular, paper is lacking an adequate computational complexity and run-time analysis with respect to the existing methods, e.g. the standard parametrization and penalty approach. A similar weakness also exists in the numerical results section, where comprehensive empirical comparisons are lacking. The supplementary material has a numerical table, which shows a very incremental improvement and is not conclusive. 5. In proving Theorem 3, the authors employ standard methods such as restricted eigenvalues. These methods are already known to extend to non-convex objective functions and does not specifically hold for the claimed convex formulation.

Correctness: Yes, both claims and empirical methodology seem to be correct.

Clarity: Yes, this paper is well written.

Relation to Prior Work: Yes, authors review the existing studies. However, I also think that more details, e.g., complexity and performance bounds, about the existing methods should be added to strengthen the claims.

Reproducibility: Yes

Additional Feedback: I have read the rebuttal which addressed some of my concerns. Please see below for my comments after the rebuttal. 1. The authors provided a new simulation result that shows better accuracy using the proposed parametrization and comparisons with non-convex penalties. Looking at these limited empirical results, it is not clear whether the claimed convexity properties of the new parametrization is critical, since the best performing method (Our-ML in the table) is based on non-convex optimization. I think the paper currently lacks a central message as the title reads "a convex optimization formulation of MV regression". These numerical comparisons (convex/non-convex loss + convex/non-convex penalty) are important, and should be discussed as part of the main paper in detail. 2. The authors also argued that convex penalties perform much worse than their non-convex versions in practice. There is a very large gap in estimation error between convex and non-convex penalties in the simulation results. However, using convex sparsity penalties (e.g. block L1) one can re-solve the problem constrained to the support to de-bias the solution. 3. The authors did not address my points regarding the restricted eigenvalue based analysis and the fact that it does not require convexity of the likelihood. I will keep my original score.


Review 4

Summary and Contributions: The paper introduces a convex optimization formulation for high-dimensional multivariate regression model under three loss functions. Various optimization algorithms are proposed to solve the associated optimization problems. The central claim is that, under a certain metric, the estimation of regression coefficients can be improved by incorporating error covariance structure, and the conditional precision matrix is also better estimated. Theoretically, the paper establishes rate of convergence in terms of vector l\infty norm and sparsistency.

Strengths: The paper introduces interesting tricks for the joint estimation of output covariance and regression coefficients. These estimators scale well, even though some issues are outlined, such as the non-SPD covariance matrices estimated.

Weaknesses: There is a logical flaw in the paper: the authors start by claiming that having convex loss is essential for the sake of estimation, but they immediately introduce non-convex penalties. The fact that the framework seems to be tied to these non-convex penalties weakens a lot the importance of the paper, almost making it an anecdote. It is unclear what the authors call high dimension, which makes many claims of the paper quite vague and hard to assess. In the experiments, 100 is supposedly high dimension ? It is a pity that empirical results are only available as supplementary material and not as part of the main paper. The claimed improvements hold for the particular metric used, but there is a circularity in the reasoning here.

Correctness: The claims could be more precise: in which situations should each estimator be used (including standard or naive estimators). This would make the paper more honest. Similarly, analyzing the behavior under more general (e.g. convex) penalties is necessary.

Clarity: Overall OK. The authors could be more precise on the relationships between B- and C- sparsity. As far as I understand, these are not strictly equivalent.

Relation to Prior Work: AFAIK prior work is acknowledged but I may have missed something

Reproducibility: Yes

Additional Feedback: 279. The fact that tau_Omega and tau_B are fixed arbitrarily is an issue: what happens if one sets them wrongly ? cross validation on lambda_1, lambda_2 should be nested. The criterion should be indicated precisely. There are some typos and grammar errors throughout the text.

[Author Response · NeurIPS 2020]

We greatly appreciate all the insightful comments and valuable suggestions on how to improve the paper.

One major concern raised by the referees centers around the use of nonconvex penalties. Here we would like to clarify our choice of nonconvex penalty over convex penalties. At a high level, the rationale for using the nonconvex penalties is that nonconvex penalties in high-dimensional models have been shown to outperform its convex counterpart both in theory and in practice in many situation [see, e.g., Fan and Li, 2001, Zhang, 2010, Shen et al., 2012, among others]. In the context of this work, we agree that convex penalties can also be used. However, the corresponding theoretical results may be less sharp in terms of both convergence rate and the conditions needed for sparsistency. Moreover, nonconvex penalty tends to perform better as compared to its convex version in practice. To empirically verify this in our context, we have performed additional simulations using convex penalties. The results are summarized in Table 1, in which "Our-ML" and "Our-ML-convex" denotes the proposed method using nonconvex and convex penalty, respectively; and "MRCE-group" denotes the MRCE method using the same convex penalty (with the old parameterization). Table 1 shows that the proposed methods with convex penalties performs much worse than the nonconvex version. More importantly, the advantages of the new reparameterization is evident by comparing our method using convex penalty ("Our-ML-convex") with MRCE using the same penalty ("MRCE-group") as they only differ in parameterization. Therefore, this means that using the proposed parameterization leads to better estimation accuracy. Finally, we would like to point out that the nonconvex penalty we used includes the convex $L_1$ penalty as a special case when $\tau = +\infty$.

Regarding to the proposed reparameterization, one referee asked about the novelty of the proposed reparameterization given that the negative loglikelihood function is convex in a classical reparameterization. Here we would like to clarify that the classical reparameterization $(B, \Theta) = (C\Omega^{1/2}, \Omega^{1/2})$ could be used if we do not want to impose sparsity-inducing penalty on $\Omega$. As one referee have pointed out, one can still impose a penalty on $B$ to retain the row-sparsity structure. However, it seems to be difficult to penalize $\Theta$ so that $\Omega$ is sparse. This is why we propose a different reparameterization so that we can impose sparsity-inducing penalties on both $C$ and $\Omega$.

Regarding to the computational details of the DC algorithm, we agree that this part was not clearly presented. We modified line 208-211 as follows

*Its key idea is to decompose the objective function into difference of two convex functions, and linearize the trailing function to obtain an upper convex approximation of the nonconvex objective. In our setting, using the DC decomposition that $p_\tau(x) = |x| - \max(|x| - \tau, 0)$, we obtain upper convex approximation of the nonconvex penalty at the previous iterate $\left(B^{(t)}, \Omega^{(t)}\right)$:*

$$p_{\tau_B}\left(\|B_{i\cdot}\|_2\right) \leq \|B_{i\cdot}\|_2 \mathbb{I}(\|B_{i\cdot}^{(t)}\|_2 \leq \tau_B) + \tau_B \mathbb{I}(\|B_{i\cdot}^{(t)}\|_2 > \tau_B)$$

$$p_{\tau_\Omega}\left(|\omega_{ij}|\right) \leq |\omega_{ij}|\mathbb{I}(|\omega_{ij}^{(t)}| \leq \tau_\Omega) + \tau_\Omega \mathbb{I}(|\omega_{ij}^{(t)}| > \tau_\Omega).$$

We have created a new revision which fixes the typos found by the reviewer and makes some minor changes to the technical proofs to make them more accessible. We are glad the reviewer appreciates the main ideas behind our paper, and in the next revision of the paper, we plan to incorporate your suggestions and the above proposed changes.

Table 1: Additional simulation results.

| $(p,q)$ | Method | Error($\widehat{C}$) | Error($\widehat{\Omega}$) | Error($\widehat{C}, \widehat{\Omega}$) | FPR($\widehat{C}$) | FNR($\widehat{C}$) |
|---|---|---|---|---|---|---|
| (200, 3) | Our-ML | .051(.028) | .031(.019) | .085(.038) | 0(.002) | 0(0) |
| | Our-ML-convex | .371(.082) | .684(.073) | .933(.058) | .434(.062) | 0(0) |
| | MRCE-group | .597(2.96) | .433(1.18) | 1.11(3.11) | .19(.117) | 0(0) |

| $(p,q)$ | Method | Error($\widehat{C}$) | Error($\widehat{\Omega}$) | Error($\widehat{C}, \widehat{\Omega}$) | FPR($\widehat{\Omega}$) | FNR($\widehat{\Omega}$) |
|---|---|---|---|---|---|---|
| (3, 200) | Our-ML | 3.01(.15) | 3.58(.203) | 6.77(.282) | .004(.001) | .004(.002) |
| | Our-ML-convex | 3.01(.15) | 8.99(.195) | 11.9(.238) | .067(.006) | 0(0) |
| | MRCE-group | 3.02(.15) | 37.1(4.8) | 4.5(4.65) | .162(.08) | .48(.255) |

# References

J. Fan and R. Li. Variable selection via nonconcave penalized likelihood and its oracle properties. *Journal of the American Statistical Association*, 96(456):1348–1360, 2001.

X. Shen, W. Pan, and Y. Zhu. Likelihood-Based Selection and Sharp Parameter Estimation. *Journal of the American Statistical Association*, 107(497):223–232, June 2012.

C. H. Zhang. Nearly unbiased variable selection under minimax concave penalty. *The Annals of Statistics*, 38(2):894–942, 2010.


[Meta-Review · NeurIPS 2020]

This paper proposes a new parametrization of the multivariate linear regression problem. It shows that under this new parametrization, it is easier to employ sparsity inducing penalty terms on the inverse covariance matrix. The paper suggests a sequential relaxation algorithm. The reviewers noted the novelty of the approach and numerous strengths. The simulation experiments (in the supplementary material) explore the method in the context of several connectivity scenarios. However, one weakness is the exploration of the performance of the model on real data scenarios. It would also be helpful to explore the sensitivity of the approach to violations of assumptions. There were other minor weaknesses around initialization, and clarity. Despite these minor weaknesses. The author's response addressed many of the concerns the reviewers raised and the discussion around the manuscript and thought it was overall a positive contribution to the community. I recommend this paper for acceptance.